# Langevin Unlearning: A New Perspective of Noisy Gradient Descent for Machine Unlearning

**Eli Chien**
Department of Electrical and Computer Engineering
Georgia Institute of Technology
Georgia, U.S.A.
`ichien6@gatech.edu`

**Haoyu Wang**
Department of Electrical and Computer Engineering
Georgia Institute of Technology
Georgia, U.S.A.
`haoyu.wang@gatech.edu`

**Ziang Chen**
Department of Mathematics
Massachusetts Institute of Technology
Massachusetts, U.S.A.
`ziang@mit.edu`

**Pan Li**
Department of Electrical and Computer Engineering
Georgia Institute of Technology
Georgia, U.S.A.
`panli@gatech.edu`

## Abstract

Machine unlearning has raised significant interest with the adoption of laws ensuring the "right to be forgotten". Researchers have provided a probabilistic notion of approximate unlearning under a similar definition of Differential Privacy (DP), where privacy is defined as statistical indistinguishability to retraining from scratch. We propose Langevin unlearning, an unlearning framework based on noisy gradient descent with privacy guarantees for approximate unlearning problems. Langevin unlearning unifies the DP learning process and the privacy-certified unlearning process with many algorithmic benefits. These include approximate certified unlearning for non-convex problems, complexity saving compared to retraining, sequential and batch unlearning for multiple unlearning requests.

## 1 Introduction

With recent demands for increased data privacy, owners of these machine learning models are responsible for fulfilling data removal requests from users. Certain laws are already in place guaranteeing the users' "Right to be Forgotten", including the European Union's General Data Protection Regulation (GDPR), the California Consumer Privacy Act (CCPA), and the Canadian Consumer Privacy Protection Act (CPPA) [1]. Merely removing user data from the training data set is insufficient, as machine learning models are known to memorize training data information [2]. It is critical to also remove the information of user data subject to removal requests from the machine learning models. This consideration gave rise to an important research direction, referred to as *machine unlearning* [3].

Naively, one may retrain the model from scratch after every data removal request to ensure a "perfect" privacy guarantee. However, it is prohibitively expensive in practice when accommodating frequent removal requests. To avoid complete retraining, various machine unlearning methods have been

38th Conference on Neural Information Processing Systems (NeurIPS 2024).

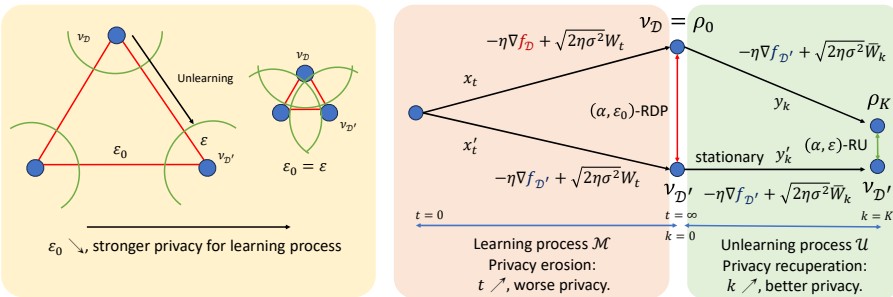

Figure 1: The geometric interpretation of relations between learning and unlearning. (Left) RDP guarantee of the learning process induces a regular polyhedron. Smaller $\varepsilon_0$ implies an "easier" unlearning problem. (Right) Learning and unlearning processes on adjacent datasets. It illustrates our main idea and results. More learning iteration gives worse privacy (**privacy erosion** [12]) while more unlearning iteration gives better privacy, which we termed this phenomenon as **privacy recuperation**.

proposed, including exact [4–6] as well as approximate approaches [1, 7–10]. Exact approaches ensure that the unlearned model would be identical to the retraining one in distribution. Approximate approaches, on the other hand, allow for slight misalignment between the unlearned model and the retraining one in distribution under a similar definition to Differential Privacy (DP) [11].

## 1.1 Our Contributions

Learning with noisy gradient methods, such as DP-SGD [13], is widely adopted for privatizing machine learning models with DP guarantee. Intuitively, a learning process with a stronger DP guarantee implies an "easier" unlearning problem as depicted in Figure 1. However, it is unclear if fine-tuning with it on the updated dataset subject to the unlearning request provides an approximate unlearning guarantee, how the DP learning guarantee affects unlearning, and computational benefit compared to retraining. In this work, we provide an affirmative answer for the empirical risk minimization problems with smooth objectives. We propose Langevin unlearning, an approximate unlearning framework based on projected noisy gradient descent (PNGD). Our core idea can be interpreted via a novel unified geometric view of the learning and unlearning processes in Figure 1, which naturally bridges DP and unlearning. Given sufficient learning iterations via the learning process $\mathcal{M}$, we first show that PNGD converges to a *unique* stationary distribution $\nu_{\mathcal{D}}$ for any dataset $\mathcal{D}$ (Theorem 3.1). Comparing $\nu_{\mathcal{D}}$ with the stationary distribution $\nu_{\mathcal{D}'}$ for any of its adjacent dataset $\mathcal{D}'$, the learning process shows Rényi DP with privacy loss[1] $\varepsilon_0$. Given a particular unlearning request $\mathcal{D} \to \mathcal{D}'$, the unlearning process $\mathcal{U}$ can be interpreted as moving from $\nu_{\mathcal{D}}$ to $\nu_{\mathcal{D}'}$ from $\varepsilon_0$-close to $\varepsilon$-close. In practice, due to the unlearning process, the unlearning privacy loss $\varepsilon$ can be set much smaller than $\varepsilon_0$, while on the other hand, a stronger initial RDP guarantee, i.e., smaller $\varepsilon_0$, allows for less unlearning iterations to achieve the desired $\varepsilon$. Besides the above DP-unlearning bridge, this framework also brings many benefits including (1) a capability of dealing with non-convex problems in theory, which to the best of our knowledge, no previous approximate unlearning framework can tackle, (2) better privacy-utility trade-off in practice compared to state-of-the-art approximate unlearning approach [8] in strongly convex settings, (3) a provably computational benefit compared with model retraining, and (4) a friendly extension to sequential and batch settings with multiple unlearning requests.

We prove the intuition in Fig. 1 formally in Theorem 3.2. We show that $K$ unlearning iterations lead to an *exponentially fast* privacy loss decay $\varepsilon \leq \exp(-\frac{1}{\alpha} \sum_{k=0}^{K-1} R_k)\varepsilon_0$, where $\alpha$ is the order of Rényi divergence and $R_k$ is the strict privacy improving rate depends on the problem settings with an iteration independent strictly positive lower bound $\bar{R} > 0$. Our result is based on convergence analysis of Langevin dynamics [15]. The sampling essence of PNGD allows for a provable unlearning guarantee for non-convex problems [16, 17]. Our characterization of $\varepsilon_0$ allows an extension of the recent results that PNGD learning satisfies Rényi DP for convex problems [12, 18, 19] to non-convex

---

[1]We refer privacy loss as two-sided Rényi divergence of two distributions, which is defined as Rényi difference in Definition 2.2. Note that it is standard to define privacy loss for unlearning as the two-sided Rényi divergence between the weight distribution of the unlearned model and the retrained model. Such a definition directly relates to the power of the strongest possible adversary for distinguishing these two cases [14].

problems as summarized in Theorem 3.3. Our key technique is to carefully track the constant of log-Sobolev inequality [20] (LSI) along the learning and unlearning processes and leverage the boundedness property of the projection step via results of [21].

Regarding the computational benefit compared to model retraining, we show iteration complexity saving by comparing two Rényi differences, the one between initialization $\nu_0$ and $\nu_{\mathcal{D}'}$, which is at least $\Omega(1)$ in the worst case versus the other one between the learning convergent distribution $\nu_{\mathcal{D}}$ and $\nu_{\mathcal{D}'}$, i.e., $\varepsilon_0$ which is shown to be $O(1/n^2)$ for a dataset of size $n$. Such a gap demonstrates that Langevin unlearning is more efficient than retraining, especially for the dataset with large $n$.

For sequential unlearning with multiple unlearning requests, we composite the privacy loss bound for single-step requests via the weak triangle inequality of Rényi divergence [22], which yields a sequential unlearning procedure that achieves privacy loss $\varepsilon$ for each request (Corollary 3.4). For batch unlearning, $\varepsilon_0$ is changed to incorporate the batch size (Theorem 3.3).

Beyond theoretical contributions, we also conduct empirical evaluation. Despite the provable order-wise improvement in $n$ compared to re-training, our current theory has a limitation by relying on some constants that are undetermined or can be only loosely determined in the non-convex setting. Therefore, we focus on logistic regression tasks for empirical evaluation. Compared with the state-of-the-art gradient-based certified approximate unlearning solution [8] that requires strong convexity, we achieve a superior privacy-utility-complexity trade-off. Although success in the convex case may not directly imply success in non-convex settings, we leave tightening these constants as a future study. For this, we discuss potential alleviation and future direction in Appendix A and 5, respectively.

## 1.2 Related Works

**Unlearning with privacy guarantees.** Prior approximate unlearning works require (strong) convexity of the objective function [1, 7, 8]. Their analysis is based on the sensitivity analysis of the *optimal parameter*. Since the optimal parameter is not even unique in the non-convex setting, it is unclear how their analysis can be generalized beyond convexity. In contrast, we show that the law of our PNGD learning process admits a *unique* stationary distribution even for *non-convex* problems. Authors of [1, 7] leverage a second-order update which requires computing Hessian inverse and thus is not scalable for high-dimensional problems. While they only require one unlearning iteration, we show in our experiment that one PNGD unlearning iteration is sufficient for strongly convex loss to achieve satisfied privacy with comparable utility to retraining as well. Neel et al. [8] leverage PGD for learning and unlearning, and achieve the privacy guarantee via publishing the final parameters with additive Gaussian noise. We show in our experiment that our Langevin unlearning strategy provides a better privacy-utility-complexity trade-off compared to this approach. Ullah et al. [5] focus on exact unlearning by leveraging variants of noisy (S)GD. Their analysis is based on total variation stability which is different from our analysis based on Rényi divergence. Also, their analysis does not directly generalize to approximate unlearning. Several works focus on extending the unlearning problems for adaptive unlearning requests [6, 9, 23]. While we focus on the non-adaptive setting, it is possible to show that Langevin unlearning is also capable of adaptive unlearning requests as we do not keep any non-private internal state. We left a rigorous discussion on this as future work. Chourasia et al. [23] also leverage Langevin dynamic analysis in their work. However, their unlearning definition is different from the standard literature as ours[2].

**Differential privacy of noisy gradient methods.** A pioneer work [24] studied the DP properties of Langevin Monte Carlo methods. Yet, they do not propose using noisy GD for general machine learning problems. A recent line of work [12, 18, 25] shows that projected noisy (S)GD training exhibits DP guarantees based on the analysis of Langevin dynamics [15, 26] under the strong convexity assumption. In the meanwhile, Altschuler et al. [19] also provided the DP guarantees for projected noisy SGD training but with analysis based on Privacy Amplification by Iteration [27] under the convexity assumption. None of these works study how PNGD can be leveraged for machine unlearning or DP guarantees for non-convex problems.

**Sampling literature.** Non-asymptotic convergence analysis for Langevin Monte Carlo has a long history [28, 29]. The seminal works [15, 24] proved non-asymptotic convergence analysis in Rényi

---

[2]Their unlearning privacy definition does not compare with retraining and they only discuss *one-side* Rényi divergence. As a result, their unlearning guarantee is less compatible with DP and cannot control both type I and II errors simultaneously against the best possible adversary [14].

divergence under strong convexity. Many works improve upon them by either working with weaker isoperimetric inequalities or different notions of convergence [30, 31]. See [26] for a more thorough review along this direction. While these works mainly focus on convergence to the unbiased limit (i.e., the limiting distribution for an infinitesimal step size), we have biased limits (i.e., the limiting distribution for a constant step size, such as our $\nu_{\mathcal{D}}$) in machine unlearning problems. Recently Altschuler et al. [32] initiated the question of studying the properties and convergence to the biased limit. Our work provides a new important application, machine unlearning, for these astonishing theoretical results in the sampling literature.

The rest of the paper is organized as follows. In Section 2, we provide preliminaries and problem setup. The theoretical results of Langevin unlearning are in Section 3. We conclude with experiments in Section 4. Due to the space limit, all proofs and future directions are deferred to Appendices.

## 2 Preliminaries and Problem Setup

We consider the empirical risk minimization (ERM) problem. Let $\mathcal{D} = \{\mathbf{d}_i\}_{i=1}^n$ be a training dataset with $n$ data points $\mathbf{d}_i$ taken from the universe $\mathcal{X}$. Let $f_{\mathcal{D}}(x) = \frac{1}{n} \sum_{i=1}^n f(x; \mathbf{d}_i)$ be the objective function. We aim to minimize with learnable parameter $x \in \mathcal{C}_R$, where $\mathcal{C}_R = \{x \in \mathbb{R}^d \mid \|x\| \leq R\}$ is a closed ball of radius $R$. We denote $\Pi_{\mathcal{C}_R} : \mathbb{R}^d \mapsto \mathcal{C}_R$ to be an orthogonal projection to $\mathcal{C}_R$. The norm $\|\cdot\|$ is standard Euclidean $\ell_2$ norm if not specified. $\mathcal{P}(\mathcal{C})$ is denoted as the set of all probability measures over a closed convex set $\mathcal{C}$. Standard definitions such as convexity can be found in Appendix C. Finally, we use $x \sim \nu$ to denote that a random variable $x$ follows the probability distribution $\nu$. To control the convergence behavior of (P)NGD, it is standard to check an isoperimetric condition known as log-Sobolev inequality [20], described as follows.

**Definition 2.1** (Log-Sobolev Inequality ($C_{\text{LSI}}$-LSI)). A probability measure $\nu \in \mathcal{P}(\mathbb{R}^d)$ is said to satisfy Logarithmic Sobolev Inequality with constant $C_{\text{LSI}}$ if

$$\forall \rho \in \mathcal{P}(\mathbb{R}^d), \;\; D_1(\rho\|\nu) \leq \frac{C_{\text{LSI}}}{2} \mathbb{E}_{x \sim \rho} \left\| \nabla \log \frac{\rho(x)}{\nu(x)} \right\|^2,$$

where $D_1(\rho\|\nu)$ is the Kullback–Leibler divergence.

### 2.1 Privacy Definition for Learning and Unlearning

We say two datasets $\mathcal{D} = \{\mathbf{d}_i\}_{i=1}^n$ and $\mathcal{D}' = \{\mathbf{d}_i'\}_{i=1}^n$ are adjacent if they "differ" only in one index $i_0 \in [n]$ so that $\mathbf{d}_i = \mathbf{d}_i'$ for all $i \neq i_0$ unless otherwise specified. Furthermore, we say two datasets $\mathcal{D}$ and $\mathcal{D}'$ are adjacent with a group size of $S \geq 1$ if they differ in at most $S$ indices. We next introduce a useful idea termed Rényi difference.

**Definition 2.2** (Rényi difference). Let $\alpha > 1$. For a pair of probability measures $\nu, \nu'$ with the same support, the $\alpha$ Rényi difference $d_\alpha(\nu, \nu')$ is defined as $d_\alpha(\nu, \nu') = \max\left(D_\alpha(\nu\|\nu'), D_\alpha(\nu'\|\nu)\right)$, where $D_\alpha(\nu\|\nu')$ is the $\alpha$ Rényi divergence $D_\alpha(\nu\|\nu')$ defined as

$$D_\alpha(\nu\|\nu') = \frac{1}{\alpha - 1} \log \left( \mathbb{E}_{x \sim \nu'} \left[ \frac{\nu(x)}{\nu'(x)} \right]^\alpha \right).$$

We are ready to introduce the formal definition of differential privacy and unlearning.

**Definition 2.3** (Rényi Differential Privacy (RDP) [22]). Let $\alpha > 1$. A randomized algorithm $\mathcal{M} : \mathcal{X}^n \mapsto \mathbb{R}^d$ satisfies $(\alpha, \varepsilon)$-RDP if for any adjacent dataset pair $\mathcal{D}, \mathcal{D}' \in \mathcal{X}^n$, the $\alpha$ Rényi difference $d_\alpha(\nu, \nu') \leq \varepsilon$, where $\mathcal{M}(\mathcal{D}) \sim \nu$ and $\mathcal{M}(\mathcal{D}') \sim \nu'$.

It is known to the literature that an $(\alpha, \varepsilon)$-RDP guarantee can be converted to the popular $(\epsilon, \delta)$-DP guarantee [11] relatively tight [22]. As a result, we will focus on establishing results with respect to $\alpha$ Rényi difference (and equivalently $\alpha$ Rényi divergence). Next, we introduce our formal privacy definition of unlearning.

**Definition 2.4** (Rényi Unlearning (RU)). Consider a randomized learning algorithm $\mathcal{M} : \mathcal{X}^n \mapsto \mathbb{R}^d$ and a randomized unlearning algorithm $\mathcal{U} : \mathbb{R}^d \times \mathcal{X}^n \times \mathcal{X}^n \mapsto \mathbb{R}^d$. We say $(\mathcal{M}, \mathcal{U})$ achieves $(\alpha, \varepsilon)$-RU if for any $\alpha > 1$ and any adjacent datasets $\mathcal{D}, \mathcal{D}'$, the $\alpha$ Rényi difference $d_\alpha(\rho, \nu') \leq \varepsilon$, where $\mathcal{U}(\mathcal{M}(\mathcal{D}), \mathcal{D}, \mathcal{D}') \sim \rho$ and $\mathcal{M}(\mathcal{D}') \sim \nu'$.

Notably, our Definition 2.4 can be converted to the standard $(\epsilon, \delta)$-unlearning definition [1, 7, 8], similar to RDP-DP conversion [22]. Since we work with the replacement definition of dataset adjacency, to "erase" a data point $\mathbf{d}_i$ we can simply replace it with any data point $\mathbf{d}'_i \in \mathcal{X}$ for the updated dataset $\mathcal{D}'$ in practice.

## 3 Langevin Unlearning: Main Results

We propose to leverage projected noisy gradient descent for our learning and unlearning algorithm $\mathcal{M}$ and $\mathcal{U}$. For $\mathcal{M}$, we propose to optimize the objective $f_{\mathcal{D}}(x)$ with PNGD:

$$x_{t+1} = \Pi_{\mathcal{C}_R} \left( x_t - \eta \nabla f_{\mathcal{D}}(x_t) + \sqrt{2\eta\sigma^2} W_t \right), \tag{1}$$

where $W_t \overset{\text{iid}}{\sim} \mathcal{N}(0, I_d)$ and $\eta, \sigma^2 > 0$ are hyperparameters of step size and noise variance respectively. The initialization $x_0$ can be chosen arbitrarily in $\mathcal{C}_R$ unless specified. We assume the learning procedure will train the model until convergence $x_\infty = \mathcal{M}(\mathcal{D})$ for simplicity, where we prove in Theorem 3.1 that the law of this learning process (1) indeed converges to a unique stationary distribution when $\nabla f_{\mathcal{D}}$ is continuous. A similar "well-trained" assumption has been also used in prior unlearning literature [1, 7] and we will discuss the case of insufficient training later. After we obtain a learned parameter $\mathcal{M}(\mathcal{D})$, an unlearning request arrives so that the training dataset changes from $\mathcal{D}$ to $\mathcal{D}'$. For the unlearning algorithm $\mathcal{U}$, we propose to fine-tune the model parameters on the new objective $f_{\mathcal{D}'}(y)$ with $K$ iterations of the same PNGD.

$$y_{k+1} = \Pi_{\mathcal{C}_R} \left( y_k - \eta \nabla f_{\mathcal{D}'}(y_k) + \sqrt{2\eta\sigma^2} \bar{W}_k \right), \tag{2}$$

where $\bar{W}_k \overset{\text{iid}}{\sim} \mathcal{N}(0, I_d)$ and $y_0 = x_\infty$, which starts from the convergent point of the learning procedure. Throughout our work, we assume $f(x; \mathbf{d})$ is $M$-Lipschitz and $L$-smooth in $x$ for any $\mathbf{d} \in \mathcal{X}$. Nevertheless, one can apply per-sample gradient clipping in (1) and (2) so that the $M$-Lipschitz assumption can be dropped. In this case, our learning and unlearning processes admit the popular DP-SGD [13] without mini-batching. For the rest of the paper, we denote $\nu_t, \rho_k$ as the laws of the processes $x_t, y_k$ respectively. Recall that we also denote the limiting distribution of the learning process (1) as $\nu_{\mathcal{D}}$ for training dataset $\mathcal{D}$.

### 3.1 Limiting Distribution and General Idea

A key component of the Langevin unlearning is the existence, uniqueness, and stationarity of the limiting distribution $\nu_{\mathcal{D}}$ of the training process. We start with proving that $\nu_{\mathcal{D}}$ exists, is unique, and is a stationary distribution. Our proof is relegated to Appendix F, which is based on showing the ergodicity of the process (1) by leveraging results in [33].

**Theorem 3.1.** *Suppose that the closed convex set $\mathcal{C}_R \subset \mathbb{R}^d$ is bounded with $\mathcal{C}_R$ having a positive Lebesgue measure and that $\nabla f_{\mathcal{D}} : \mathcal{C}_R \to \mathbb{R}^d$ is continuous. The Markov chain $\{x_t\}$ in (1) admits a unique invariant probability measure $\nu_{\mathcal{D}}$ on the Borel $\sigma$-algebra of $\mathcal{C}_R$. Furthermore, for any $x \in \mathcal{C}_R$, the distribution of $x_t$ conditioned on $x_0 = x$ converges weakly to $\nu_{\mathcal{D}}$ as $t \to \infty$.*

If $\mathcal{M}$ is known to be $(\alpha, \varepsilon_0)$-RDP for a $\alpha > 1$, by definition we know that for all adjacent dataset $\mathcal{D}, \mathcal{D}'$, $d_\alpha(\nu_{\mathcal{D}}, \nu_{\mathcal{D}'}) \leq \varepsilon_0$. In the space of $\mathcal{P}(\mathcal{C}_R)$, this RDP guarantee gives a "regular polyhedron", where vertices are $\nu_{\mathcal{D}}, \nu_{\mathcal{D}'}$ and all adjacent vertices are of "lengths" $\varepsilon_0$ at most in Rényi difference. We caveat that Rényi difference is not a metric but the idea of the regular polyhedron is useful conceptually. As a result, the RDP guarantee of the learning process controls the "distance" between distribution induced from adjacent dataset $\mathcal{D}$ and $\mathcal{D}'$. Once we finish the learning process, we receive an unlearning request so that our dataset changes from $\mathcal{D}$ to an adjacent dataset $\mathcal{D}'$. We need to move from $\nu_{\mathcal{D}}$ to $\nu_{\mathcal{D}'}$ at least $\varepsilon$ close for a $(\alpha, \varepsilon)$-RU guarantee. Intuitively, if the initial RDP guarantee is stronger (i.e., $\varepsilon_0$ is smaller), unlearning becomes "easier" at the cost of larger noise. When $\varepsilon_0 = \varepsilon$, we automatically achieve $(\alpha, \varepsilon)$-RU without any unlearning update. One of our main contributions is to characterize how many PNGD unlearning iteration is needed to reduce $d_\alpha(\rho_k, \nu_{\mathcal{D}'})$ from $\varepsilon_0$ to $\varepsilon$, where $\rho_0 = \nu_{\mathcal{D}}$.

For the unlearning process, note that the initial Rényi difference between $\rho_0, \nu_{\mathcal{D}'}$ is provided by the RDP guarantees of the learning process. As a result, we are left to characterize the convergence of

the process $y_k$ to its stationary distribution $\nu_{\mathcal{D}'}$ in Rényi difference (Theorem 3.2). Since the privacy loss $\varepsilon$ gradually decays with respect to unlearning iterations, we refer to this phenomenon as **privacy recuperation**. This is in contrast to the learning process, where prior work [12] has shown the worse privacy loss $\varepsilon_0$ with respect to learning iterations and refers to that phenomenon as **privacy erosion**.

## 3.2 Unlearning Guarantees

Our first Theorem shows that $(\mathcal{M}, \mathcal{U})$ achieves $(\alpha, \varepsilon)$-RU, where $\varepsilon$ decays monotonically in $K$ unlearning iterations starting from $\varepsilon_0$, condition on $\mathcal{M}$ being $(\alpha, \varepsilon_0)$-RDP. We provide the proof sketch in Appendix G.1 and formal proofs are deferred to Appendix G.2.

**Theorem 3.2** (RU guarantee of PNGD unlearning). *Assume for all $\mathcal{D} \in \mathcal{X}^n$, $f_{\mathcal{D}}$ is $L$-smooth, $M$-Lipschitz and $\nu_{\mathcal{D}}$ satisfies $C_{LSI}$-LSI. Let the learning process follow the PNGD update* (1). *Given $\mathcal{M}$ is $(\alpha, \varepsilon_0)$-RDP and $y_0 = x_\infty = \mathcal{M}(\mathcal{D})$, for $\alpha > 1$, the output of the $K^{th}$ unlearning iteration along* (2) *(i.e., $y_K$) achieves $(\alpha, \varepsilon)$-RU, where $\varepsilon \le \exp\left(-\frac{1}{\alpha} \sum_{k=0}^{K-1} R_k\right) \varepsilon_0$ and $R_k > 0$ depends on the problem settings specified as follows:*

*1) For a general non-convex $f_{\mathcal{D}}$, we have $R_k = \frac{1}{2}\left(\frac{1}{((1+\eta L)^2 C_k)^2} - \frac{1}{((1+\eta L)^2 C_k + 2\eta\sigma^2)^2}\right)$, where $C_{k+1} \le \min((1+\eta L)^2 C_k + 2\eta\sigma^2, \tilde{C})$, $\tilde{C} = 6(4(R+\eta M)^2 + 2\eta\sigma^2)\exp(\frac{4(R+\eta M)^2}{2\eta\sigma^2})$, where $C_0 = C_{LSI}$ and $R$ is the radius of the projected set $\mathcal{C}_R$.*

*2) Suppose $f_{\mathcal{D}}$ is convex. By choosing $\eta \le \frac{2}{L}$, we have $R_k = \frac{1}{2}\left(\frac{1}{(C_k)^2} - \frac{1}{(C_k + 2\eta\sigma^2)^2}\right)$, where $C_{k+1} \le \min(C_k + 2\eta\sigma^2, \tilde{C})$.*

*3) Suppose $f_{\mathcal{D}}$ is $m$-strongly convex. Let $\frac{\sigma^2}{m} < C_{LSI}$ and choosing $\eta \le \min(\frac{2}{m}(1 - \frac{\sigma^2}{mC_{LSI}}), \frac{1}{L})$. Then, $R_k = \frac{2\sigma^2\eta}{C_{LSI}}$.*

Note that $R_k$ can be interpreted as the strict privacy improving rate at step $k$ and $C_k$ is the LSI constant upper bound of distribution of $y_k$. The above theorem states that fine-tuning with PNGD can decrease the privacy loss $d_\alpha(\rho_K, \nu_{\mathcal{D}'})$ *exponentially fast* with the unlearning iteration $K$. This is because $R_k$ is lower bounded away from 0 by a constant, thanks to the iteration independent upper bound on $C_k$. Stronger assumptions on the objective function $f_{\mathcal{D}}$ lead to a better rate, which implies fewer unlearning iterations are needed to achieve the same RU guarantee. There are several remarks for our Theorem 3.2. First, note that the result is *dimension-free*, which is favorable for problems with many parameters to be learned. Second, note that the $M$-Lipschitzness assumption can be dropped by clipping the gradient to norm $M$ in the PNGD update (1) and (2) instead. As a result, our Theorem 3.2 applies to neural networks with smooth activation functions in theory. Finally, our result gives an *upper bound* on the LSI constants along the unlearning process (i.e., $C_k$) which may be improved with more advanced analysis. We note that the exponential dependence in $R$ for the bound of $C_k$ can be loose. It is possible to have a better constant with either more structural assumptions or working with different isoperimetric inequalities such as (weak) Poicaré inequality [31]. A more detailed discussion is in Appendix 5.

**Initial RDP guarantees and LSI constant.** Since Theorem 3.2 relies on $\mathcal{M}$ being $(\alpha, \varepsilon_0)$-RDP and the $\nu_{\mathcal{D}}$ satisfies LSI, the theorem below provides such results for the learning process, where the formal proof is relegated to Appendix H.

**Theorem 3.3** (RDP guarantee of PNGD learning). *Assume $f(\cdot; \mathbf{d})$ be $L$-smooth and $M$-Lipschitz for all $\mathbf{d} \in \mathcal{X}$. Also assume that the initialization of PNGD* (1) *satisfies $C_0$-LSI. Then the learning process* (1) *is $(\alpha, \varepsilon_0^{(S)})$-RDP of group size $S \ge 1$ at $T^{th}$ iteration with*

$$\varepsilon_0^{(S)} \le \frac{2\alpha\eta S^2 M^2}{\sigma^2 n^2} \sum_{t=1}^{T} \prod_{t'=0}^{t-1} (1 + \frac{\eta\sigma^2}{C_{t',1}})^{-1},$$

*where $C_{t,1} \le \min\left((1+\eta L)^2 C_t + \eta\sigma^2, \bar{C}\right)$, $\bar{C} = 6(4(R+\eta M)^2 + \eta\sigma^2)\exp(\frac{4(R+\eta M)^2}{\eta\sigma^2})$ and $C_{t+1} \le \min\left(C_{t,1} + \eta\sigma^2, \bar{C}\right)$. Furthermore, $\nu_t$ satisfies $C_t$-LSI.*

*When we additionally assume $f(\cdot; \mathbf{d})$ is convex, by choosing $\eta \le \frac{2}{L}$ the same result hold with $C_{t,1} \le \min\left(C_t + \eta\sigma^2, \bar{C}\right)$.*

*When we additionally assume $f(\cdot; \mathbf{d})$ is $m$-strongly convex, by choosing $0 < \eta \leq \min(\frac{2}{m}(1 - \frac{\sigma^2}{mC_0}), \frac{1}{L})$ with a constant $C_0 > \frac{\sigma^2}{m}$, we have $\varepsilon_0^{(S)} \leq \frac{4\alpha S^2 M^2}{m\sigma^2 n^2}(1 - \exp(-m\eta T))$. Furthermore, $\nu_t$ satisfies $C_0$-LSI for all $t \geq 0$.*

Note that any initialization $x_0 \in \mathcal{C}_R$ can be viewed as sampling from $\mathcal{N}(x_0, cI_d)$ with $c \to 0$, which corresponds to $C_0$-LSI for any $C_0 > 0$. By taking $T \to \infty$, Theorem 3.3 provides the initial $(\alpha, \varepsilon_0)$-RDP guarantee and the LSI constant needed in Theorem 3.2. Since there is an iteration-independent upper bound for $C_{t,1}$, one can show that $\varepsilon_0 \leq \frac{2\alpha\eta S^2 M^2}{\sigma^2 n^2 c}$ for some $T$-independent constant $c \in (0, 1)$ due to the finiteness of geometric series. Similar to our discussion for Theorem 3.2, the bound of $C_t$ may be loose and it is possible to further improve the LSI constant analysis. The goal of our results is to demonstrate that it is possible to derive (finite) RDP and (arbitrarily small) RU guarantees even for general non-convex problems.

Nevertheless, for the $m$-strongly convex case we have $\varepsilon_0^{(S)} \leq \frac{4\alpha S^2 M^2}{m\sigma^2 n^2}$ for all $T > 0$, including $T \to \infty$. It shows that indeed the current learned distribution $\nu_{\mathcal{D}}$ is close to the retraining distribution $\nu_{\mathcal{D}'}$ for $n$ sufficiently large. This also leads to the computational benefit of Langevin unlearning compared to retraining from scratch, which we discuss below. On the other hand, we show by experiments in Section 4 that our results provide a superior privacy-utility-complexity trade-off for the strongly convex case compared to existing approximate unlearning approaches.

**The computational benefit compared to retraining.** While our Theorems 3.2 and 3.3 together provide the privacy guarantee of Langevin unlearning, it is critical to check if our approach provides a computational benefit compared to retraining from scratch as well. Let $\nu_0$ be the (data-independent) initialization distribution of the learning process. Intuitively, starting with $\nu_{\mathcal{D}}$ instead of $\nu_0$ (i.e., retraining) should converge faster to $\nu_{\mathcal{D}'}$, since $d_\alpha(\nu_{\mathcal{D}}, \nu_{\mathcal{D}'}) \leq \varepsilon_0$ is likely to be much smaller than $d_\alpha(\nu_0, \nu_{\mathcal{D}'})$. Thus, our Langevin unlearning needs less iterations than retraining for most cases, except for a corner case when $\nu_0$ is already close to $\nu_{\mathcal{D}}$. From Theorem 3.2 we know that the number of PNGD iterations we need to approach $\varepsilon$-close in $d_\alpha$ to the target distribution $\nu_{\mathcal{D}'}$ is roughly $O(\log(\frac{\varepsilon_I}{\varepsilon}))$, where $\varepsilon_I$ is the Rényi difference between the initial distribution and the target distribution $\nu_{\mathcal{D}'}$. From Theorem 3.3, we know that the initial Rényi difference of Langevin unlearning is at most $\varepsilon_0 = O(1/n^2)$ for any datasets $\mathcal{D}, \mathcal{D}'$ and any smooth Lipchitz loss. In contrast, even if both the target distribution $\nu_{\mathcal{D}'}$ and the initialization of retraining $\nu_0$ are Gaussian distributions with the same variance but mean difference $\Omega(1)$, their Rényi difference is $\Omega(1)$ [22]. As a result, computational saving offered by Langevin unlearning is significant for sufficiently large $n$. A more thorough discussion is in Appendix E.

### 3.3 Empirical Aspects of Langevin Unlearning

**Insufficient training.** While our theorem assumes the learning process runs until convergence, this assumption can be relaxed by the geometric view of Langevin unlearning. Assume the learning process $\mathcal{M}(\mathcal{D}) \sim \nu_T$ terminate with finite step $T$ instead and we only have $d_\alpha(\nu_T, \nu_{\mathcal{D}}) \leq \varepsilon_T(\alpha)$ for all possible $\mathcal{D} \in \mathcal{X}^n$. One can still apply the weak triangle inequality of Rényi divergence [22] twice to bound $d_\alpha(\rho_k, \nu_T')$ with $d_{4\alpha}(\rho_k, \nu_{\mathcal{D}'})$, $\varepsilon_T(2\alpha)$, and $\varepsilon_T(4\alpha)$ with additional factors $(\alpha - 0.5)/(\alpha - 1)$ and $(2\alpha - 0.5)/(2\alpha - 1)$. In practice, it is reasonable to require the model parameters to be sufficiently trained so that $\varepsilon_T$ is negligible and a tighter weak triangle inequality can be employed.

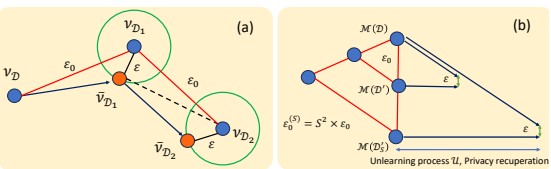

Figure 2: Illustration of (a) sequential unlearning and (b) batch unlearning. For sequential unlearning, we can leverage the weak triangle inequality of Rényi divergence to connect all the error terms. For batch unlearning, only the initial RDP guarantee changes with a general group size. Notably, unlearning more samples at once implies $\varepsilon_0$ being larger (Theorem 3.3), and thus we need more unlearning iteration to recuperate the privacy loss to a desired $\varepsilon$.

**Sequential and batch unlearning.** Langevin unlearning naturally supports sequential and batch unlearning for unlearning multiple data points thanks to our geometric view of the unlearning problem, see Figure 2 for a pictorial example. For sequential unlearning, we show that fine-tuning the current model parameters on the updated datasets for sequential $S \geq 1$ unlearning requests can achieve $(\alpha, \varepsilon)$-RU simultaneously. The formal proof is deferred to Appendix I.

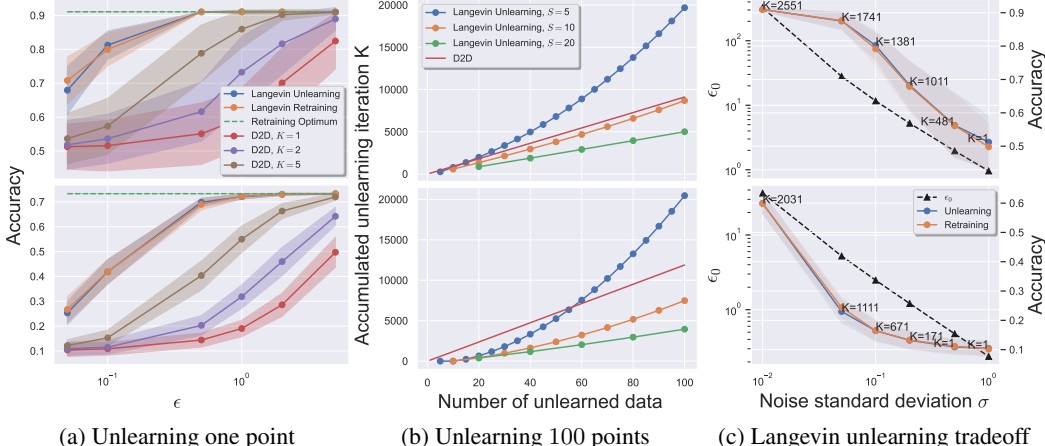

Figure 3: Main experiments, where the top and bottom rows are for MNIST and CIFAR10 respectively. (a) Compare to D2D for unlearning one point using limited unlearning iteration. This demonstrates the privacy-utility ($\epsilon$-accuracy) tradeoff under the fixed unlearning complexity (K). For Langevin unlearning, we use only $K = 1$ unlearning iterations. For D2D, we allow it not only to use $K = 1, 2, 5$ unlearning iterations but also to keep the non-private internal state information. (b) Compare to D2D for unlearning 100 points, where all methods achieve $(\epsilon, 1/n)$-unlearning guarantee with $\epsilon = 1$. For Langevin unlearning, we vary different unlearning batch sizes $S$ and combine them with the sequential unlearning result. For D2D, we do not allow it to keep the non-private internal state information in this experiment so that there is an inherent lower bound on the unlearning iterations per unlearning request. (c) A detailed investigation of the utility-complexity trade-off of Langevin unlearning with unlearning $S = 100$ points at once under the fixed privacy constraint $\epsilon = 1$. For each $\sigma$, we report the corresponding $\epsilon_0$ (black dash line) for the initial $(\epsilon_0, 1/n)$-DP guarantee and the utility after unlearning to $\epsilon = 1$.

**Corollary 3.4** (Sequential unlearning). *Assume the unlearning requests arrive sequentially such that our dataset changes from $\mathcal{D}_0 \to \mathcal{D}_1 \to \ldots \to \mathcal{D}_S$, where $\mathcal{D}_s, \mathcal{D}_{s+1}$ are adjacent. Let $y_k^{(s)}$ be the unlearned parameters for the $s^{th}$ unlearning request with $k$ unlearning update following (2) on $\mathcal{D}_s$ and $y_0^{(s+1)} = y_{K_s}^{(s)} \sim \bar{\nu}_{\mathcal{D}_s}$, where $y_0^{(1)} = x_\infty$ and $K_s$ is the unlearning steps for the $s^{th}$ unlearning request. Suppose we have achieved $(\alpha, \varepsilon^{(s)}(\alpha))$-RU for the $s^{th}$ unlearning request, the learning process (1) is $(\alpha, \varepsilon_0(\alpha))$-RDP and $\bar{\nu}_{\mathcal{D}_s}$ satisfies $C_{LSI}$-LSI, we achieve $(\alpha, \varepsilon^{(s+1)}(\alpha))$-RU for the $(s+1)^{th}$ unlearning request as well, where*

$$\varepsilon^{(s+1)}(\alpha) \leq \exp(-\frac{1}{\alpha} \sum_{k=0}^{K_{s+1}-1} R_k) \times \frac{\alpha - 1/2}{\alpha - 1} \left( \varepsilon_0(2\alpha) + \varepsilon^{(s)}(2\alpha) \right),$$

$\varepsilon^{(0)}(\alpha) = 0 \, \forall \alpha > 1$ *and $R_k$ are defined in Theorem 3.2.*

As a result, one can leverage Corollary 3.4 to recursively determine needed unlearning iterations for each sequential unlearning request. For the batch unlearning setting, it only affects the initial Rényi difference in Theorem 3.2. We can simply adopt Theorem 3.3 with a group size of $S \geq 1$ for the RDP guarantees of the learning process $\varepsilon_0^{(S)}$.

**Utility-privacy-efficiency trade-off.** An interesting aspect of the Langevin unlearning is its strong connection to the initial RDP guarantee. From Theorem 3.3, we know that increasing $\sigma$ leads to smaller Rényi difference $\varepsilon_0$ and thus better unlearning efficiency. However, this intuitively is at the cost of the utility of $\nu_{\mathcal{D}}$, see for example the discussion in Section 5 of [12] under the strong convexity assumption. To achieve the same $(\alpha, \varepsilon)$-RU guarantee, one can either ensure smaller $\varepsilon_0$ at the cost of worst utility or run more unlearning iterations at the cost of unlearning efficiency. We investigate how utility trade-off with privacy and unlearning complexity empirically in Section 4.

# 4 Experiments

*Benchmark datasets.* We consider logistic regression with $\ell_2$ regularization. We focus on this strongly convex setting since the non-convex unlearning bound in Theorem 3.2 currently is not tight enough to be applied in practice due to its exponential dependence on various hyperparameters. However, we emphasize its significant theoretical implication due to the lack of a certified non-convex approximate unlearning framework in previous studies. Meanwhile, the existing baseline approach [8] also only applies to strongly convex problems. We conduct experiments on MNIST [34] and CIFAR10 [35], which contain 11,982 and 50,000 training instances respectively. We follow the setting of [7] to distinguish digits 3 and 8 for MNIST so that the problem is a binary classification. For the CIFAR10 dataset, we use all classes and leverage the last layer of the public ResNet18 [36] embedding as the data features, which follows the public feature extractor setting of [7]. Experiments on additional datasets [37] are deferred to Appendix M and our code is publicly available[3].

*Baseline methods.* Our baseline methods include Delete-to-Descent (D2D) [8], the state-of-the-art gradient-based approximate unlearning method, and retraining from scratch using PNGD. For D2D, we leverage Theorem 9 and 28 in [8] for privacy accounting depending on whether we allow D2D to have an internal non-private state. Note that allowing an internal non-private state provides a weaker notion of privacy guarantee [8] and our Langevin unlearning by default does not require it. We include those theorems for D2D and a detailed explanation of its possible non-privacy internal state in Appendix N for completeness. All experimental details can be found in Appendix M, including how to convert $(\alpha, \varepsilon)$-RU to the standard $(\epsilon, \delta)$-unlearning guarantee. Throughout this section, we choose $\delta = 1/n$ for each dataset and require all tested unlearning approaches to achieve $(\epsilon, \delta)$-unlearning with different $\epsilon$. We report test accuracy for all experiments as the utility metric. For the initialization, we sample from Gaussian distribution with mean 1000. This simulates the case that the initial distribution is in a reasonable distance away from the convergent distribution $\nu_{\mathcal{D}}$. We set the learning iteration $T = 10,000$ to ensure all approaches converge. For Langevin unlearning, we leverage Theorems 3.2, 3.3 and Corollay 3.4 for privacy accounting under different settings. All results are averaged over 100 independent trials with standard deviation reported as shades in all figures.

**Unlearning one data point with $K = 1$ iteration.** We first consider the setting of unlearning one data point using only $K = 1$ unlearning iteration for both Langevin unlearning and D2D (Figure 3a). Since D2D cannot achieve a privacy guarantee with only 1 unlearning iteration without a non-private internal state, we allow D2D to have it in this experiment. Even in this case, our Langevin unlearning significantly outperforms D2D in utility for $\epsilon$ from 0.1 to 5 under the same unlearning complexity ($K = 1$), but also achieves similar accuracy to retraining from scratch. Since retraining requires $T = 10,000$ PNGD iterations, Langevin unlearning is indeed much more efficient. We also show that D2D can achieve better utility at the cost of a larger unlearning iteration $K = 2, 5$. Our Langevin unlearning exhibits both smaller unlearning complexity and better utility compared to D2D.

**Unlearning multiple data points.** We now consider the scenario of unlearning 100 data points, where the results are in Figure 3b. We let all methods achieve the same $(1, 1/n)$-unlearning guarantee for a fair comparison. Since D2D only supports sequential unlearning, we directly apply its sequential unlearning results [8]. Also, we do not allow D2D to have an internal non-private state in this experiment for a fair comparison. On the other hand, since Langevin unlearning supports both sequential and batch unlearning, we vary the number of points per unlearning request $S = 5, 10, 20$ and report the accumulated unlearning iterations for $\sigma = 0.03$. All methods achieve a similar utility, with an accuracy of roughly 0.9 and 0.98 for MNIST and CIFAR10 respectively. Langevine unlearning can achieve a significantly better unlearning complexity compared to D2D if one allows for a larger unlearning batch size. For instance, when we are allowed to unlearn $S = 20$ points at once, Langevine unlearning saves 40% unlearning iteration compared to D2D. Nevertheless, we note that due to the use of weak triangle inequality of Rényi divergence in our analysis, Langevin unlearning can be more

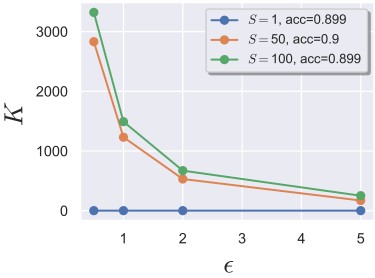

Figure 4: Trade-off between privacy ($\epsilon$), unlearning complexity ($K$), and the number of points to be unlearned ($S$) in the batch unlearning setting for MNIST. We fix $\sigma = 0.03$ so that $K$ can be determined given $(\epsilon, S)$.

---

[3]https://github.com/Graph-COM/Langevin_unlearning

expensive in complexity compared to D2D when one only allows for unlearning a small batch of points (i.e, $S = 5$). We leave the improvement in this direction as the future work.

**Privacy-utility-complexity trade-off.** We further examine the inherent privacy-utility-complexity trade-off provided by our Langevin unlearning with two experiments. In the first experiment, we aim to achieve $(\epsilon, 1/n)$-unlearning guarantee with $\epsilon = 1$ for batch unlearning of 100 points. We vary the choice of $\sigma$ from 0.01 to 1.0. A smaller $\sigma$ leads to a worse initial $\epsilon_0$ and thus requires more unlearning iteration $K$ to recuperate it to $\epsilon = 1$. It is interesting to see that even if we choose a small $\sigma$ so that the initial $(\epsilon_0, 1/n)$-DP guarantee is extremely weak (i.e, $\epsilon_0 \approx 100$ for $\sigma = 0.05$), our unlearning iteration can recuperate $\epsilon_0$ to $\epsilon = 1.0$ efficiently. On the other hand, a larger $\sigma$ leads to a worse utility which is the inherent privacy-utility-complexity trade-off of Langevin unlearning. The results are illustrated in Figure 3c. Compared to retraining until convergence ($T = 10,000$), we achieve a similar utility but with much lower unlearning complexity with $K$ roughly up to 2500.

In the second experiment, we investigate the effect of the number of points to be unlearned $S$ in the batch unlearning setting. In Figure 4, we can see that both larger $S$ and smaller $\epsilon$ will require more unlearning iterations $K$. It is worth noting that the resulting utility does not change significantly, whereas Langevin unlearning always archives a similar utility compared to retraining (see Figure 5a in Appendix M). Retraining requires $T = 10,000$ PNGD iterations which is significantly larger than the required unlearning iteration $K$ even for $\epsilon = 0.5$. We have shown that Langevin unlearning is a promising unlearning solution.

# 5    Conclusions and Future Directions

We propose Langevin unlearning based on noisy gradient descent with privacy guarantees for approximate unlearning problems. It unifies the DP learning process and the privacy-certified unlearning process with many algorithmic benefits such as applicability to non-convex problems and multiple unlearning requests. Below we discuss several future directions for Langevin unlearning.

**Extension to projected noisy stochastic gradient descent.** It is straightforward to extend our analysis to the projected noisy SGD case. There are two possibilities for the SGD setting: 1) randomly partition the indices $[n]$ into a sequence of mini-batches, then fix this sequence for all the learning and unlearning process [18]; 2) randomly draw a mini-batch for each update [19, 25]. The analysis of [18] can be combined with our LSI constant analysis for RU guarantees, similar to the proof of our Theorem 3.2. Unfortunately, the analysis [25] may lead to an extra large LSI constant in the intermediate step even if $R$ is small. We refer interested readers to Appendix C of [18] for a detailed discussion. The technical difficulty here is to provide a tight analysis of the LSI constant for a mixture of distributions, where each of them corresponds to a possible choice of mini-batch. The analysis of [19] is based on privacy amplification by iteration, which does not directly generalize to the non-convex cases. In our companion work [38], we exploit it not only to establish a better unlearning result but also to enable the mini-batch setting under the convexity assumption. It is currently an open problem whether a matching result can be established via Langevin dynamic analysis as well.

**Better convergence rate.** While it is already exciting that Langevin dynamic analysis leads to formal unlearning algorithms and guarantees even for general non-convex problems in theory, the potential of this direction for a *practical* plug-and-play unlearning solution is even more interesting. Several promising future directions can significantly improve the convergence rate and the unlearning efficiency. Developing a better LSI constant bound under additional structural assumptions for the non-convex problems is the most straightforward one. Another direction is to work with (weak) Poincaré inequality instead. While a weaker tail assumption leads to slower convergence [31], the corresponding (weak) Poincaré constant may be more tightly tracked. Finally, while we only discuss the noisy GD which corresponds to Langevin Monte Carlo, some other advanced samplers are off-the-shelf including the Metropolis-Hastings filter [39] and Hamiltonian Monte Carlo [40]. We hope our work motivates further collaborations among the sampling and privacy communities and pushes the boundaries of learning and unlearning with privacy guarantees.

## Acknowledgments and Disclosure of Funding

The authors thank Sinho Chewi, Wei-Ning Chen, and Ayush Sekhari for the helpful discussions. E. Chien, H. Wang and P. Li are supported by NSF awards OAC-2117997 and JPMC faculty award.

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

## A Limitations

While our Langevin unlearning provides the first approximate unlearning solution for non-convex problems, it is still not practical enough as we have stated in the main text. One limitation of our unlearning guarantees (Theorem 3.2) is that the privacy bound for non-convex problems is not tight and can potentially improved via advanced analysis, where some possible directions are discussed in the future directions below. Currently, it is still only applicable in practice for strongly convex problems as we have demonstrated in our experiment section if a privacy guarantee is required. Nevertheless, recent studies have shown that a $(\epsilon, \delta)$-DP model provides strong empirical privacy against popular attacks such as membership inference attacks even if $\epsilon \approx 10^8$ [41]. We conjecture a similar phenomenon exists for Langevin unlearning, which allows Langevin unlearning to defend against membership inference attacks for non-convex models as well in practice with a similar $\epsilon$ scale. We leave this empirical study as our important future work.

## B Broader Impact

Our work study the theoretical unlearning guarantees of projected noisy gradient descent algorithm for convex problems. We believe our work is a foundational research and does not have a direct path to any negative applications.

## C Standard Definitions

Let $f : \mathbb{R}^d \mapsto \mathbb{R}$ be a mapping. We define smoothness, Lipschitzsness, and strong convexity as follows:

$$L\text{-smooth: } \forall\, x, y \in \mathbb{R}^d,\ \|\nabla f(x) - \nabla f(y)\| \leq L\|x - y\| \tag{3}$$

$$m\text{-strongly convex: } \forall\, x, y \in \mathbb{R}^d,\ \langle x - y, \nabla f(x) - \nabla f(y)\rangle \geq m\|x - y\|^2 \tag{4}$$

$$M\text{-Lipschitzs: } \forall\, x, y \in \mathbb{R}^d,\ \|f(x) - f(y)\| \leq M\|x - y\|. \tag{5}$$

Furthermore, we say $f$ is convex means it is $0$-strongly convex.

## D Additional Related Works

**Bayesian Unlearning.** Due to the relation between Langevin Monte Carlo and Bayes learning approaches, our Langevin unlearning is also loosely related to the Bayesian unlearning literature. See [42–44] for a series of empirical results. Along this line of work, [45] is the only one that provides a certain unlearning guarantee in terms of KL divergence. However, they only provide a bound for one direction of KL (similar to $D_1(\rho_k \| \nu'_{\mathcal{D}})$) which makes it fail to be directly connected to the differential privacy. Note that it is crucial to ensure the bidirectional bound for KL or Rényi divergence for the purpose of privacy. Otherwise, we cannot ensure the sufficiently large type I and type II errors of the best possible attacker in membership inference attack [14]. Also, it is essential to have a (relatively) tight conversion to DP, where the general $\alpha$ order in Rényi divergence is crucial.

## E Detailed Discussion on Computational Benefit Against Retraining

In this section, we provide a more detailed discussion of the computational benefit of Langevin unlearning against retraining from scratch. We start our discussion under strongly convex assumption and then explain the non-convex case. Let us consider the case $f(x; \mathbf{d})$ is $m$-strongly convex, $L$-smooth and $M$-Lipschitz in $x$ for all $\mathbf{d} \in \mathcal{X}$. Also, assume the initialization distribution $\nu_0 = \mathcal{N}(\tilde{x}_0, \frac{2\sigma^2}{m} I_d)$ for some $\tilde{x}_0 \in \mathcal{C}_R$. In this case, from Theorem 3.2 we know that running $T$ PNGD learning iteration (1), we have

$$d_\alpha(\nu_T, \nu_{\mathcal{D}'}) \leq \exp(-\frac{2\sigma^2 \eta T}{\alpha C_{\text{LSI}}}) d_\alpha(\nu_0, \nu_{\mathcal{D}'}). \tag{6}$$

Note that by Theorem 3.3, we know that $C_{\text{LSI}} = \frac{2\sigma^2}{m}$ by our choice of $\nu_0$ for an appropriate step size $\eta \leq \min(\frac{1}{m}, \frac{1}{L})$. As a result, in order to be $\varepsilon$ close to $\nu_{\mathcal{D}'}$, we need $\frac{\alpha}{m\eta} \log(\frac{d_\alpha(\nu_0, \nu_{\mathcal{D}'})}{\varepsilon})$ retraining

iteration. On the other hand, for Langevin unlearning we need $\frac{\alpha}{m\eta}\log(\frac{\varepsilon_0}{\varepsilon})$, where $\varepsilon_0 \leq \frac{4\alpha M^2}{m\sigma^2 n^2}$. As a result, Langevin unlearning the computational saving for Langevin unlearning against retraining is

$$\frac{\alpha}{m\eta}\log(\frac{d_\alpha(\nu_0,\nu_{\mathcal{D}'})}{\varepsilon}) - \frac{\alpha}{m\eta}\log(\frac{\varepsilon_0}{\varepsilon}) = \frac{\alpha}{m\eta}\log(\frac{d_\alpha(\nu_0,\nu_{\mathcal{D}'})}{\varepsilon_0}) \geq \frac{\alpha}{m\eta}\log(\frac{m\sigma^2 n^2 \times d_\alpha(\nu_0,\nu_{\mathcal{D}'})}{4\alpha M^2}). \tag{7}$$

Clearly, this saving depends on $\nu_{\mathcal{D}'}$. In some rare cases, $\nu_0$ might accidentally be close to $\nu_{\mathcal{D}'}$ so that retraining is more efficient. However, even if $\nu_{\mathcal{D}'} = \mathcal{N}(x^\star(\mathcal{D}'), \frac{2\sigma^2}{m}I_d)$, we have $d_\alpha(\nu_0,\nu_{\mathcal{D}'}) = \frac{\alpha m\|\tilde{x}_0 - x^\star(\mathcal{D}')\|}{4\sigma^2}$. That is, even if we know the target distribution is Gaussian and choose the initialization to have the same variance, the corresponding Rényi difference is $\Omega(1)$ for $\|\tilde{x}_0 - x^\star(\mathcal{D}')\| = \Omega(1)$. As a result, if we uniformly at random sample $\tilde{x}_0$ from $\mathcal{C}_R$, we have $\|\tilde{x}_0 - x^\star(\mathcal{D}')\| \geq 1$ with probability at least $1 - \frac{1}{R^d}$. Plug this into the lower bound above we obtain a data-independent lower bound on the computational savings with probability at least $1 - \frac{1}{R^d}$ as follows

$$\frac{\alpha}{m\eta}\log(\frac{d_\alpha(\nu_0,\nu_{\mathcal{D}'})}{\varepsilon}) - \frac{\alpha}{m\eta}\log(\frac{\varepsilon_0}{\varepsilon}) \geq \frac{\alpha}{m\eta}\log(\frac{m\sigma^2 n^2 \times d_\alpha(\nu_0,\nu_{\mathcal{D}'})}{4\alpha M^2}) \tag{8}$$

$$= \frac{\alpha}{m\eta}\log(\frac{m^2 n^2 \|\tilde{x}_0 - x^\star(\mathcal{D}')\|}{16M^2}) \geq \frac{\alpha}{m\eta}\log(\frac{m^2 n^2}{16M^2}). \tag{9}$$

Here we can see that for larger problem size $n$, our computational benefit is more significant.

For the non-convex case, note that the convergence rate $R_k$ in Theorem 3.2 will vary and depend on the LSI constant of $\nu_0$ and $\nu_{\mathcal{D}}$ in general. This makes it hard to have a direct characterization of the computational benefit against retraining. To simplify the situation, we assume the convergence rate $R_k$ is a constant $\bar{R} > 0$ that is independent of $n, k$. In this case, the computational saving can be characterized as

$$\frac{\alpha}{\bar{R}}\log(\frac{d_\alpha(\nu_0,\nu_{\mathcal{D}'})}{\varepsilon_0}). \tag{10}$$

In this case, one can still leverage Theorem 3.3 to provide an upper bound on $\varepsilon_0$, yet the obtained bound can be weak due to the inaccurate estimate of LSI constants for the non-convex case. Instead, we propose to use unbiased limits $\tilde{\nu}_{\mathcal{D}}$ to approximate the biased limit $\nu_{\mathcal{D}}$ for a rough estimate instead, since $D_\alpha(\nu_{\mathcal{D}}\|\tilde{\nu}_{\mathcal{D}}) \to 0$ as $\eta \to 0$ [26]. From standard sampling literature [15], we know that $\tilde{\nu}_{\mathcal{D}} \propto \exp(-\frac{f_{\mathcal{D}}}{\sigma^2})$. We provide the following result for bounding $d_\alpha(\tilde{\nu}_{\mathcal{D}}, \tilde{\nu}_{\mathcal{D}'})$.

**Proposition E.1.** *Let $\tilde{\nu}_{\mathcal{D}} \propto \exp(-f_{\mathcal{D}})$. Assume $|f(x; \mathbf{d}) - f(x; \mathbf{d}')| \leq F$ for all $x \in \mathbb{R}^d$ and $\mathbf{d}, \mathbf{d}' \in \mathcal{X}$. Then $d_\alpha(\tilde{\nu}_{\mathcal{D}}, \tilde{\nu}_{\mathcal{D}'}) \leq \frac{2F}{n}$ for any adjacent dataset $\mathcal{D}, \mathcal{D}'$ and $\alpha > 1$.*

As a result, we know that $\varepsilon_0$ is roughly at most $\frac{2F}{\sigma^2 n}$ when the step size $\eta$ is sufficiently small. Thus when $d_\alpha(\nu_0,\nu_{\mathcal{D}'}) = \Omega(1)$, we Langevin unlearning save $\Omega(\log(n))$ PNGD iterations.

## F Proof of Theorem 3.1: Convergence of PNGD

**Theorem.** *Suppose that the closed convex set $\mathcal{C}_R \subset \mathbb{R}^d$ is bounded with $Leb(\mathcal{C}_R) > 0$ where $Leb$ denotes the Lebesgue measure and that $\nabla f_{\mathcal{D}} : \mathcal{C}_R \to \mathbb{R}^d$ is continuous. The Markov chain $\{x_t\}$ in (1) admits a unique invariant probability measure $\nu_{\mathcal{D}}$ on $\mathcal{B}(\mathcal{C}_R)$ that is the Borel $\sigma$-algebra of $\mathcal{C}_R$. Furthermore, for any $x \in \mathcal{C}_R$, the distribution of $x_t$ conditioned on $x_0 = x$ converges weakly to $\nu_{\mathcal{D}}$ as $t \to \infty$.*

In this section, we prove that the learning process (1) with general closed convex set $\mathcal{C}$ that is restated as follows for the reader's convenience,

$$x_{t+1} = \Pi_{\mathcal{C}}\left(x_t - \eta \nabla f_{\mathcal{D}}(x_t) + \sqrt{2\eta\sigma^2}W_t\right), \tag{11}$$

will converge to an invariant probability measure. One observation is that (11) is a Markov chain and some ergodicity results can be applied.

**Proposition F.1.** *Suppose that the closed convex set $\mathcal{C} \subset \mathbb{R}^d$ is bounded with $Leb(\mathcal{C}) > 0$ where Leb denotes the Lebesgue measure and that $\nabla f_{\mathcal{D}} : \mathcal{C} \to \mathbb{R}^d$ is continuous. Then the Markov chain $\{x_t\}$ defined by* (11) *admits a unique invariant measure (up to constant multiples) on $\mathcal{B}(\mathcal{C})$ that is the Borel $\sigma$-algebra of $\mathcal{C}$.*

*Proof.* This proposition is a direct application of results from [33]. According to Proposition 10.4.2 in [33], it suffices to verify that $\{x_t\}$ is recurrent and strongly aperiodic.

1. *Recurrency.* Thanks to the Gaussian noise $W_t$, $\{x_t\}$ is Leb-irreducible, i.e., it holds for any $x \in \mathcal{C}$ and any $A \in \mathcal{B}(\mathcal{C})$ with $Leb(A) > 0$ that

$$L(x, A) := \mathbb{P}(\tau_A < +\infty \mid x_0 = x) > 0,$$

where $\tau_A = \inf\{t \geq 0 : x_t \in A\}$ is the stopping time. Therefore, there exists a Borel probability measure $\psi$ such that that $\{x_t\}$ is $\psi$-irreducible and $\psi$ is maximal in the sense of Proposition 4.2.2 in [33]. Consider any $A \in \mathcal{B}(\mathcal{C})$ with $\psi(A) > 0$. Since $\{x_t\}$ is $\psi$-irreducible, one has $L(x, A) = \mathbb{P}(\tau_A < +\infty \mid x_0 = x) > 0$ for all $x \in \mathcal{C}$. This implies that there exists $T \geq 0$, $\delta > 0$, and $B \in \mathcal{B}(\mathcal{C})$ with $Leb(B) > 0$, such that $\mathbb{P}(x_T \in A \mid x_0 = x) \geq \delta$, $\forall x \in B$. Therefore, one can conclude for any $x \in \mathcal{C}$ that

$$U(x, A) := \sum_{t=0}^{\infty} \mathbb{P}(x_t \in A \mid x_0 = x)$$

$$\geq \sum_{t=1}^{\infty} \mathbb{P}(x_{t+T} \in A \mid x_t \in B, x_0 = x) \cdot \mathbb{P}(x_t \in B \mid x_0 = x)$$

$$\geq \sum_{t=1}^{\infty} \delta \cdot \inf_{y \in \mathcal{C}} \mathbb{P}(x_t \in B \mid x_{t-1} = y)$$

$$= +\infty,$$

where we used the fact that $\inf_{y \in \mathcal{C}} \mathbb{P}(x_t \in B \mid x_{t-1} = y) = \inf_{y \in \mathcal{C}} \mathbb{P}(x_1 \in B \mid x_0 = y) > 0$ that is implies by $Leb(B) > 0$ and the boundedness of $\mathcal{C}$ and $\nabla f_{\mathcal{D}}(\mathcal{C})$. Let us remark that we actually have compact $\nabla f_{\mathcal{D}}(\mathcal{C})$ since $\mathcal{C}$ is compact and $\nabla f_{\mathcal{D}}$ is continuous. The arguments above verify that $\{x_t\}$ is recurrent (see Section 8.2.3 in [33] for definition).

2. *Strong aperiodicity.* Since $\mathcal{C}$ and $\nabla f_{\mathcal{D}}(\mathcal{C})$ are bounded and the density of $W_t$ has a uniform positive lower bound on any bounded domain, there exists a non-zero multiple of the Lebesgue measure, say $\nu_1$, satisfying that

$$\mathbb{P}(x_1 \in A \mid x_0 = x) \geq \nu_1(A), \quad \forall x \in \mathcal{C}, \ A \in \mathcal{B}(\mathcal{C}).$$

Then $\{x_t\}$ is strongly aperiodic by the equation above and $\nu_1(\mathcal{C}) > 0$ (see Section 5.4.3 in [33] for definition).

The proof is hence completed. □

**Theorem F.2.** *Under the same assumptions as in Proposition F.1, the Markov chain $\{x_t\}$ admits a unique invariant probability measure $\nu_{\mathcal{D}}$ on $\mathcal{B}(\mathcal{C})$. Furthermore, for any $x \in \mathcal{C}$, the distribution of $x_t$ generated by* (11) *conditioned on $x_0 = x$ converges weakly to $\nu_{\mathcal{D}}$ as $t \to \infty$.*

*Proof.* It has been proved in Proposition F.1 that $\{x_t\}$ is strongly aperiodic and recurrent with an invariant measure. Consider any $A \in \mathcal{B}(\mathcal{C})$ with $\psi(A) > 0$ and use the same settings and notations as in the proof of Proposition F.1. There exists $T \geq 0$, $\delta > 0$, and $B \in \mathcal{B}(\mathcal{C})$ with $Leb(B) > 0$, such that $\mathbb{P}(x_T \in A \mid x_0 = x) \geq \delta$, $\forall x \in B$. This implies that for any $t \geq 0$ and any $x \in \mathcal{C}$,

$$\mathbb{P}(x_{t+T+1} \in A \mid x_t = x) = \mathbb{P}(x_{T+1} \in A \mid x_0 = x) \geq \mathbb{P}(x_{T+1} \in A \mid x_1 \in B, x_0 = x) \cdot \mathbb{P}(x_1 \in B \mid x_0 = x) \geq \epsilon,$$

where

$$\epsilon = \delta \cdot \inf_{y \in \mathcal{C}} \mathbb{P}(x_1 \in B \mid x_0 = y) > 0,$$

which then leads to

$$Q(x, A) := \mathbb{P}(x_t \in A, \text{ infinitely often}) = +\infty.$$

This verifies that the chain $\{x_t\}$ is Harris recurrent (see Section 9 in [33] for definition). It can be further derived that for any $x \in \mathcal{C}$,

$$\mathbb{E}(\tau_A \mid x_0 = x) = \sum_{t=1}^{\infty} \mathbb{P}(\tau_A \geq t \mid x_0 = x) \leq (T+1) \sum_{k=0}^{\infty} \mathbb{P}(\tau_A > (T+1)k \mid x_0 = x)$$

$$\leq (T+1) \sum_{k=1}^{\infty} (1-\epsilon)^k < +\infty.$$

The bound above is uniform for all $x \in \mathcal{C}$ and this implies that $\mathcal{C}$ is a regular set of $\{x_t\}$ (see Section 11 in [33] for definition). Finally, one can apply Theorem 13.0.1 in [33] to conclude that there exists a unique invariant probability measure $\nu_{\mathcal{D}}$ on $\mathcal{B}(\mathcal{C})$ and that the distribution of $x_t$ converges weakly to $\nu_{\mathcal{D}}$ conditioned on $x_0 = x$ for any $x \in \mathcal{C}$. $\qquad \square$

## G    Proof of Theorem 3.2

### G.1    Proof Sketch of Theorem 3.2

Given $\mathcal{M}$ is $(\alpha, \varepsilon_0)$-RDP, we aim to show the upper bound of $d_\alpha(\rho_k, \nu_{\mathcal{D}'})$ decays in $k$ starting from $\varepsilon_0$ at $k = 0$. As a warm-up, we start with the strongly convex case. The analysis is inspired by [15] and [24] and formal proof can be found in Appendix G.2. Roughly speaking, we characterize how both $\alpha$ Rényi divergence $D_\alpha(\rho_k || \nu_{\mathcal{D}'})$ and $D_\alpha(\nu_{\mathcal{D}'} || \rho_k)$ decay, given $\nu_{\mathcal{D}'}$ and $\rho_k$ satisfy LSI condition for some constants. Standard sampling literature only focuses on the part $D_\alpha(\rho_k || \nu_{\mathcal{D}'})$ (i.e., Lemma 8 in [15]), where $\nu_{\mathcal{D}'}$ satisfies LSI implies exponential decay in Rényi divergence. The other direction is necessary for meaningful privacy guarantee but more challenging as one to carefully track the LSI constant of $\rho_k$ for all $k \geq 0$. We prove the following lemma for such LSI constant characterization along the unlearning process, which specializes results of [26] to the PNGD update.

**Lemma G.1** (LSI constant characterization). *Consider the following PNGD update for a closed convex set $\mathcal{C}$:*

$$x_{k,1} = h(x_k), \; x_{k,2} = x_{k,1} + \sigma W_k, \; x_{k+1} = \Pi_{\mathcal{C}}(x_{k,2}),$$

*where $h$ is any $M$-Lipschitz map $\mathbb{R}^d \mapsto \mathbb{R}^d$, $W_k \sim \mathcal{N}(0, I_d)$ independent of anything before step $k$, and $\Pi_{\mathcal{C}}$ is the projection onto a closed convex set $\mathcal{C}$. Let $\mu_{k,1}, \mu_{k,2}$ and $\mu_k$ be the distribution of $x_{k,1}$, $x_{k,2}$ and $x_k$ respectively. Then we have the following LSI constant characterization of this process. 1) If $\mu_k$ satisfies $c$-LSI, $\mu_{k,1}$ satisfies $M^2 c$-LSI. 2) If $\mu_{k,1}$ satisfies $c$-LSI, $\mu_{k,2}$ satisfies $(c + \sigma^2)$-LSI. 3) If $\mu_{k,2}$ satisfies $c$-LSI, $\mu_{k+1}$ satisfies $c$-LSI.*

By leveraging Lemma G.1, we can characterize the LSI constant for all $\rho_k$. One key step is to characterize the Lipschitz constant of the gradient update $h(x) = x - \eta \nabla f(x)$. From Lemma 2.2 in [32] we know if $f$ is $m$-strongly convex, $L$-smooth and $\eta \leq \frac{1}{L}$, then $h$ is $(1 - \eta m)$-Lipschitz. Let $\rho_k$ satisfy $C_k$-LSI, Lemma G.1 leads to the recursion expression $C_{k+1} \leq (1 - \eta m)^2 C_k + 2\eta\sigma^2$, $C_0 = C_{\text{LSI}}$. By choosing $\eta$ satisfying $0 < \eta \leq \min(\frac{2}{m}(1 - \frac{\sigma^2}{mC_{\text{LSI}}}), \frac{1}{L})$ and the assumption $\frac{\sigma^2}{m} < C_{\text{LSI}}$, $C_k$ is non-increasing and thus $\rho_k$ is $C_{\text{LSI}}$-LSI for all $k \geq 0$. As a result, the decay of both $D_\alpha(\rho_k || \nu_{\mathcal{D}'})$ and $D_\alpha(\nu_{\mathcal{D}'} || \rho_k)$ can be shown.

**Beyond strong convexity.** To extend beyond strong convexity, one may naively apply Lemma G.1 for convex and non-convex settings. Unfortunately, both cases lead to monotonically increasing LSI constant $C_k$. As a result, given an $\varepsilon_0$, proving to achieve an arbitrarily small $\varepsilon$ is challenging even with $K \to \infty$ since the LSI constant may be unbounded. More specifically, if $f$ is convex and $\eta \leq \frac{2}{L}$, then $h(x) = x - \eta \nabla f(x)$ is 1-Lipschitz. If $f$ is $L$-smooth only, the map $h$ is $(1 + \eta L)$-Lipschitz. Applying Lemma G.1 leads to the recursions on $C_k$. For the convex case, we have $C_{k+1} \leq C_k + 2\eta\sigma^2$. For the non-convex case, we have $C_{k+1} \leq (1 + \eta L)^2 C_k + 2\eta\sigma^2$.

One of our contributions is to demonstrate that $C_k$ has a universal upper bound which is independent of the number of iterations. Hence, the exponential decay in Rényi difference still holds. The key is to leverage the geometry of $\mathcal{C}_R$ to establish an LSI upper bound that is independent of $k$ using the result of [21], which has not been explored in the prior privacy literature [12, 18, 25].

**Lemma G.2** (Corollary 1 in [21]). *Let $\mu$ be a probability measure supported on $\mathcal{C}_R$ for some $R \geq 0$. Then, for each $\xi \geq 0$, $\mu * \mathcal{N}(0, \xi I_d)$ satisfy $C$-LSI with constant $C \leq 6(4R^2 + \xi)\exp(\frac{4R^2}{\xi})$.*

[19] also leverage the geometry of $\mathcal{C}_R$ for the DP guarantee of learning with projected noisy (S)GD, but their analysis follows privacy amplification by iteration [27] and still require convexity. Our result demonstrates the potential of Langevin dynamic analysis for unlearning guarantees of non-convex problems.

## G.2 Formal Proof

We will start with the proof for the strongly convex case and then extend it for convex and non-convex cases. As indicated in our sketch of proof, there are two main parts of our proof. The first is to characterize the decay in Rényi divergence between two processes $y_k, y'_k$ under LSI conditions. The second is to track the LSI constant of $y_k, y'_k$ throughout the unlearning process. The analysis is a modification of the proof of Lemma 8 in [15].

We first define some useful quantities and list all technical lemmas that we need to proof. For $\alpha > 0$ and any two probability distribution $\rho, \nu$ with the same support, define

$$F_\alpha(\rho; \nu) = \mathbb{E}_\nu[(\frac{\rho}{\nu})^\alpha] = \int \nu(x)(\frac{\rho}{\nu})^\alpha(x)\, dx. \tag{12}$$

$$G_\alpha(\rho; \nu) = \mathbb{E}_\nu[(\frac{\rho}{\nu})^\alpha \|\nabla \log \frac{\rho}{\nu}\|^2] = \mathbb{E}_\nu[(\frac{\rho}{\nu})^{\alpha-2}\|\nabla \frac{\rho}{\nu}\|^2] = \frac{4}{\alpha^2}\mathbb{E}_\nu[\|\nabla(\frac{\rho}{\nu})^{\alpha/2}\|^2]. \tag{13}$$

Note that $D_\alpha(\rho||\nu) = \frac{1}{\alpha-1}\log F_\alpha(\rho; \nu)$ by definition and $G_\alpha(\rho; \nu)$ is known as the Rényi Information, where the limit $\alpha = 1$ recovers the relative Fisher information [15]. Now we introduce all the technical lemmas we need. The first is data-processing inequality for Rényi divergence, which is the Lemma 2.6 in [32]. The second and third lemmas are based on results in [15]. We note again that we use the definition of LSI in [26], where the LSI constant is reciprocal to those defined in [15].

**Lemma G.3** (Data-processing inequality for Rényi divergence [32]). *For any $\alpha \geq 1$, any function $h : \mathbb{R}^d \mapsto \mathbb{R}^d$ and any distribution $\mu, \nu$ with support on $\mathbb{R}^d$,*

$$D_\alpha(h_\#\mu||h_\#\nu) \leq D_\alpha(\mu||\nu). \tag{14}$$

**Lemma G.4** (Lemma 18 in [15], with customized variance). *For any probability distribution $\rho_0$, $\nu_0$ and for any $t \geq 0$, let $\rho_t = \rho_0 * \mathcal{N}(0, 2t\sigma^2 I_d)$ and $\nu_t = \nu_0 * \mathcal{N}(0, 2t\sigma^2 I_d)$. Then for all $\alpha > 0$ we have*

$$\frac{d}{dt}D_\alpha(\rho_t||\nu_t) = -\alpha\sigma^2\frac{G_\alpha(\rho_t; \nu_t)}{F_\alpha(\rho_t; \nu_t)}. \tag{15}$$

**Lemma G.5** (Low bound of G-F ratio, Lemma 5 [15]). *Suppose $\nu$ satisfy $C_{LSI}$-LSI. Let $\alpha \geq 1$. For all probability distribution $\rho$ we have*

$$\frac{G_\alpha(\rho; \nu)}{F_\alpha(\rho; \nu)} \geq \frac{2}{\alpha^2 C_{LSI}}D_\alpha(\rho||\nu). \tag{16}$$

Now we are ready to prove Theorem 3.2 under strong convexity assumption.

*Proof.* For brevity and to make our proof succinct, we will only prove the harder direction $D_\alpha(\nu_{\mathcal{D}'}||\rho_k)$. The proof of the other direction is not only simpler (due to $\nu_{\mathcal{D}'}$ being the stationary distribution), but also the same analysis applies.

First, let us consider two processes:

$$y_{k+1} = \Pi_\mathcal{C}\left(y_k - \eta\nabla f_{\mathcal{D}'}(y_k) + \sqrt{2\eta\sigma^2}W_k\right), \text{where } y_0 \sim \rho_0 = \nu_\mathcal{D} \tag{17}$$

$$y'_{k+1} = \Pi_\mathcal{C}\left(y'_k - \eta\nabla f_{\mathcal{D}'}(y'_k) + \sqrt{2\eta\sigma^2}W_k\right), \text{where } y'_0 \sim \nu_{\mathcal{D}'}. \tag{18}$$

Note that $y_k$ is the process we would have during the unlearning process and $y'_k$ is an auxiliary process. Let $\rho_{k,1}, \rho_{k,2}, \rho_k$ be the probability distribution of $y_{k,1}, y_{k,2}, y_k$ respectively, where

$$y_{k,1} = y_k - \eta\nabla f_{\mathcal{D}'}(y_k), \; y_{k,2} = y_{k,1} + \sqrt{2\eta\sigma^2}W_k, \; y_{k+1} = \Pi_\mathcal{C}(y_{k,2}). \tag{19}$$

Similarly, let $\rho'_{k,1}, \rho'_{k,2}, \rho'_k$ be the probability distribution of $y'_{k,1}, y'_{k,2}, y'_k$ respectively. By definition $\nu_{\mathcal{D}'}$ is the stationary distribution of this process (in fact, both), we know that $\rho'_k = \nu_{\mathcal{D}'}$ for all $k \geq 0$.

Also, without loss of generality, we assume $\rho_k$ satisfies $C_k$-LSI for some value $C_k$ to be determined. Notably by assumption we have $C_0 = C_{\text{LSI}}$.

Observe that the gradient update $h(y) = y - \eta \nabla f_{\mathcal{D}'}(y)$ is a $(1 - \eta m)$-Lipschitz map for $f_{\mathcal{D}'}$ being $L$-smooth and $m$-strongly convex due to Lemma 2.2 in [32] when $\eta \leq \frac{1}{L}$. By Lemma G.1 we know that $\rho_{k,1}$ satisfies $((1 - \eta m)^2 C_k)$-LSI. Next, by Lemma G.3 we have

$$D_\alpha(\rho'_{k,1}||\rho_{k,1}) = D_\alpha(h_\# \rho'_k||h_\# \rho_k) \leq D_\alpha(\rho'_k||\rho_k) = D_\alpha(\nu_{\mathcal{D}'}||\rho_k). \tag{20}$$

Next, consider $\rho_{k,1,t} = \rho_{k,1} * \mathcal{N}(0, 2t\sigma^2 I_d)$ and $\rho'_{k,1,t} = \rho_{k,1} * \mathcal{N}(0, 2t\sigma^2 I_d)$ for $t \in [0, \eta]$. Clearly, $\rho_{k,1,\eta} = \rho_{k,2}$ and $\rho'_{k,1,\eta} = \rho'_{k,2}$. By Lemma G.4 we have

$$\frac{d}{dt} D_\alpha(\rho'_{k,1,t}||\rho_{k,1,t}) = -\sigma^2 \alpha \frac{G_\alpha(\rho'_{k,1,t}; \rho_{k,1,t})}{F_\alpha(\rho'_{k,1,t}; \rho_{k,1,t})}. \tag{21}$$

By Lemma G.1, we know that $\rho_{k,1,t}$ satisfies $((1 - \eta m)^2 C_k + 2\eta \sigma^2)$-LSI for all $t \leq \eta$. By the choice $\eta \leq \frac{2}{m}(1 - \frac{\sigma^2}{mC_k})$, we know that

$$(1 - \eta m)^2 C_k + 2\eta \sigma^2 \leq C_k. \tag{22}$$

Clearly, this would require $\frac{\sigma^2}{m} < C_k$ for $\eta > 0$. Then by Lemma G.5, we have

$$\frac{G_\alpha(\rho'_{k,1,t}; \rho_{k,1,t})}{F_\alpha(\rho'_{k,1,t}; \rho_{k,1,t})} \geq \frac{2}{\alpha^2 C_k} D_\alpha(\rho'_{k,1,t}||\rho_{k,1,t}). \tag{23}$$

This would imply

$$\frac{d}{dt} D_\alpha(\rho'_{k,1,t}||\rho_{k,1,t}) \leq -\frac{2\sigma^2}{\alpha C_k} D_\alpha(\rho'_{k,1,t}||\rho_{k,1,t}). \tag{24}$$

By Gronwall's inequality [46], integrating over $t \in [0, \eta]$ gives

$$D_\alpha(\rho'_{k,2}||\rho_{k,2}) \leq \exp(-\frac{2\sigma^2 \eta}{\alpha C_k}) D_\alpha(\rho'_{k,1}||\rho_{k,1}). \tag{25}$$

Apply Lemma G.3 for the mapping $\Pi_{\mathcal{C}}$, we have

$$D_\alpha(\rho'_{k+1}||\rho_{k+1}) \leq D_\alpha(\rho'_{k,2}||\rho_{k,2}). \tag{26}$$

Note that by Lemma G.1, we have also shown that $\rho_{k+1}$ is $C_k$-LSI. This implies $\rho_k$ is $C_0$-LSI, where $C_0 = C_{\text{LSI}}$ by our assumption. Combining all results and the fact that $\nu_{\mathcal{D}'}$ is the stationary distribution, we have

$$D_\alpha(\nu_{\mathcal{D}'}||\rho_{k+1}) \leq \exp(-\frac{2\sigma^2 \eta}{\alpha C_{\text{LSI}}}) D_\alpha(\nu_{\mathcal{D}'}||\rho_k). \tag{27}$$

Iterating this over $k$ we complete the proof. $\qquad\square$

The proof beyond strong convexity is similar, except the characterization of $C_k$ is different. As we mentioned in the main text, without strong convexity we can only prove an upper bound of the LSI constant that grows monotonically with respect to the iterations. To prevent a diverging LSI constant, we leverage the boundedness of the projected set $\mathcal{C}_R$ to establish an iteration-independent bound for the LSI constant. Below we give the proof of Theorem 3.2 without the strong convexity assumption.

*Proof.* As before, we will only prove the decay of the direction $D_\alpha(\nu_{\mathcal{D}'}||\rho_k)$, since it is more challenging. We again assume $\rho_k$ is $C_k$-LSI, where $C_0 = C_{\text{LSI}}$ by our assumption. First, due to [47] we know that the map $h(y) = y - \eta \nabla f_{\mathcal{D}'}(y)$ is $(1 + \eta L)$-Lipschitz for $f_{\mathcal{D}'}$ being $L$-smooth. By Lemma G.1 we know that $\rho_{k,1}$ satisfies $((1 + \eta L)^2 C_k)$-LSI. Next, by Lemma G.3 we have

$$D_\alpha(\rho'_{k,1}||\rho_{k,1}) \leq D_\alpha(\rho'_k||\rho_k) = D_\alpha(\mathcal{D}'||\rho_k). \tag{28}$$

Next, by Lemma G.4 we have

$$\frac{d}{dt} D_\alpha(\rho'_{k,1,t}||\rho_{k,1,t}) = -\sigma^2 \alpha \frac{G_\alpha(\rho'_{k,1,t}; \rho_{k,1,t})}{F_\alpha(\rho'_{k,1,t}; \rho_{k,1,t})}. \tag{29}$$

Note that by Lemma G.1, $\rho_{k,1,t}$ satisfies $((1+\eta L)^2 C_k + 2t\sigma^2)$-LSI. Then by Lemma G.5, we have

$$\frac{d}{dt} D_\alpha(\rho'_{k,1,t}||\rho_{k,1,t}) \leq -\frac{2\sigma^2}{\alpha((1+\eta L)^2 C_k + 2t\sigma^2)} D_\alpha(\rho'_{k,1,t}||\rho_{k,1,t}). \tag{30}$$

By Gronwall's inequality [46], integrating over $t \in [0, \eta]$ gives

$$D_\alpha(\rho'_{k,2}||\rho_{k,2}) \leq \exp(-\int_{t=0}^{\eta} \frac{2\sigma^2}{\alpha((1+\eta L)^2 C_k + 2t\sigma^2)} dt) D_\alpha(\rho'_{k,1}||\rho_{k,1}). \tag{31}$$

Note the calculation

$$\int_{t=0}^{\eta} \frac{2\sigma^2}{\alpha((1+\eta L)^2 C_k + 2t\sigma^2)} dt = \int_{t=0}^{\eta} \frac{d(2\sigma^2 t)}{\alpha((1+\eta L)^2 C_k + 2t\sigma^2)} \tag{32}$$

$$= \frac{1}{2\alpha} \left( \frac{1}{((1+\eta L)^2 C_k)^2} - \frac{1}{((1+\eta L)^2 C_k + 2\eta\sigma^2)^2} \right) \tag{33}$$

By applying Lemma G.3 for the projection operator and combining all results we have

$$D_\alpha(\rho'_{k+1}||\rho_{k+1}) \leq \exp(-\frac{1}{2\alpha} \left( \frac{1}{((1+\eta L)^2 C_k)^2} - \frac{1}{((1+\eta L)^2 C_k + 2\eta\sigma^2)^2} \right)) D_\alpha(\rho'_k||\rho_k). \tag{34}$$

Iterate this over $K$ steps, we have

$$D_\alpha(\nu_{\mathcal{D}'}||\rho_K) \leq \exp(-\frac{1}{2\alpha} \sum_{k=0}^{K-1} \frac{1}{2\alpha} \left( \frac{1}{((1+\eta L)^2 C_k)^2} - \frac{1}{((1+\eta L)^2 C_k + 2\eta\sigma^2)^2} \right)) D_\alpha(\nu_{\mathcal{D}'}||\rho_0) \tag{35}$$

$$= \exp(-\frac{1}{2\alpha} \sum_{k=0}^{K-1} \frac{1}{2\alpha} \left( \frac{1}{((1+\eta L)^2 C_k)^2} - \frac{1}{((1+\eta L)^2 C_k + 2\eta\sigma^2)^2} \right)) D_\alpha(\nu_{\mathcal{D}'}||\rho_0). \tag{36}$$

To complete the proof, we establish the recursion relation of $C_k$. If $f$ is convex and $\eta \leq \frac{2}{L}$, then $h(x) = x - \eta\nabla f(x)$ is 1-Lipschitz. If $f$ is $L$-smooth only, the map $h$ is $(1+\eta L)$-Lipschitz. Applying Lemma G.1 leads to the following recursions on $C_k$

$$\text{Convex: } C_{k+1} \leq C_k + 2\eta\sigma^2 \quad \text{Non-convex: } C_{k+1} \leq (1+\eta L)^2 C_k + 2\eta\sigma^2. \tag{37}$$

On the other hand, the Corollary 1 in [21] states the following result.

**Lemma G.6** (Corollary 1 in [21])**.** *Let $\mu$ be a probability measure on $\mathbb{R}^d$ supported on $\mathcal{C}_R$ for some $R \geq 0$. Then, for each $t \geq 0$, $\mu * \mathcal{N}(0, tI_d)$ satisfy C-LSI with constant*

$$C \leq 6(4R^2 + t) \exp(\frac{4R^2}{t}). \tag{38}$$

Now, consider the following PNGD process similar to Lemma G.1

$$x_{k,1} = h(x_k), \ x_{k,2} = x_{k,1} + 2\eta\sigma^2 W_k, \ x_{k+1} = \Pi_{\mathcal{C}_R}(x_{k,2}),$$

where $h(x) = x - \eta\nabla f_{\mathcal{D}}(x)$ and $W_k \sim \mathcal{N}(0, I_d)$ as before. Clearly, due to the projection $\Pi_{\mathcal{C}_R}$ we know that $\mu_k$ is supported on $\mathcal{C}_R$. By assumption that $f_{\mathcal{D}}$ is $M$-Lipschitz, we know that $\|f_{\mathcal{D}}(x)\| \leq M$ and thus $\mu_{k,1}$ is supported on $\mathcal{C}_{R+\eta M}$. By applying Lemma G.6 we know that $\mu_{k,2}$ satisfies LSI with constant upper bounded by

$$6(4(R + \eta M)^2 + 2\eta\sigma^2) \exp(\frac{4(R + \eta M)^2}{2\eta\sigma^2}). \tag{39}$$

Finally, by Lemma G.1 we know that the projection $\Pi_{\mathcal{C}_R}$ does not increase the LSI constant so that the same LSI constant upper bound holds for all $\mu_k$. Combining with our previous recursive result we complete the proof.

If we further have that $f_{\mathcal{D}'}$ being convex, then by Lemma 3.7 in [47] we know that when $\eta \leq \frac{2}{L}$ the gradient map is 1-Lipchitz. As a result, the factor $(1+\eta L)^2$ can be reduced to 1. $\qquad\square$

# H Proof of Theorem 3.3

The proof is mainly modified from the analysis of [18] with our LSI constant analysis. First, we list the all needed notations and technical lemmas adopted from [18]. Let us start with the PNGD process with training dataset $\mathcal{D}$ and $\mathcal{D}'$ as before

$$x_{t+1} = \Pi_{\mathcal{C}_R}\left(x_t - \eta\nabla f_{\mathcal{D}}(x_t) + \sqrt{2\eta\sigma^2}W_t\right), \tag{40}$$

$$x'_{t+1} = \Pi_{\mathcal{C}_R}\left(x'_t - \eta\nabla f_{\mathcal{D}'}(x'_t) + \sqrt{2\eta\sigma^2}W_t\right), \; W_t \overset{\text{iid}}{\sim} \mathcal{N}(0, I_d), \tag{41}$$

For each iteration, the above update is equivalent to the following two steps:

$$x_{t,1} = x_t - \eta\nabla f_{\mathcal{D}}(x_t) + \sqrt{\eta\sigma^2}W_t, \; x_{t+1} = \Pi_{\mathcal{C}_R}\left(x_{t,1} + \sqrt{\eta\sigma^2}W_t\right). \tag{42}$$

That is, it can be decomposed into another noisy GD update followed by a small additive noise with projection. Let $\nu_t, \nu_{t,1}, \nu'_t, \nu'_{t,1}$ be the law of $x_t, x_{t,1}, x'_t, x'_{t,1}$ respectively. Finally, we introduce the following technical lemma from [18] specialized to PNGD case.

**Lemma H.1** (Simplification of Lemma 3.2 in [18]). *For any $\xi_t, \xi'_t \in \mathcal{P}(\mathbb{R}^d)$ that both satisfy $C_{t,1}$-LSI, then we have*

$$\frac{D_\alpha(\xi_t * \mathcal{N}(0, \eta\sigma^2 I_d)||\xi'_t * \mathcal{N}(0, \eta\sigma^2 I_d))}{\alpha} \leq \frac{D_{\alpha'}(\xi_t||\xi'_t)}{\alpha'}(1 + \frac{\eta\sigma^2}{C_{t,1}})^{-1}, \; \alpha' = \frac{\alpha - 1}{1 + \frac{\eta\sigma^2}{C_{t,1}}} + 1. \tag{43}$$

The proof is an application of Lemma G.5 but with the integral involving time-dependent LSI constant. Now we are ready to prove our Theorem 3.3.

*Proof.* We first provide a full characterization of the LSI constant of $\nu_t, \nu_{t,1}$ for all $k \geq 0$, assuming $\nu_0$ is $C_0$-LSI to be chosen later. Let us denote the LSI constant of $\nu_t, \nu_{t,1}$ to be $C_t, C_{t,1}$ respectively.

By Lemma G.1, when $f_{\mathcal{D}}$ is $L$-smooth we have that

$$C_{t,1} \leq (1 + \eta L)^2 C_t + \eta\sigma^2, \quad C_{t+1} \leq C_{t,1} + \eta\sigma^2. \tag{44}$$

Similarly, by leveraging the same analysis in the proof of Theorem 3.2, using Lemma G.6 with the assumption that $f_{\mathcal{D}}$ is $M$-Lipschitz gives the following $k$ independent bound

$$C_{t,1} \leq 6(4(R + \eta M)^2 + \eta\sigma^2)\exp(\frac{4(R + \eta M)^2}{\eta\sigma^2}), \tag{45}$$

$$C_{t+1} \leq 6(4(R + \eta M)^2 + 2\eta\sigma^2)\exp(\frac{4(R + \eta M)^2}{2\eta\sigma^2}). \tag{46}$$

Now we establish the one iteration bound on the Rényi divergence. By composition theorem for RDP (and equivalently Rényi divergence) [22] and the assumption that $f_{\mathcal{D}}, f_{\mathcal{D}'}$ are $M$-Lipschitz, we have

$$\frac{D_\alpha(\nu_{t,1}||\nu'_{t,1})}{\alpha} \leq \frac{D_\alpha(\nu_t||\nu'_t)}{\alpha} + \frac{2\eta S^2 M^2}{\sigma^2 n^2}. \tag{47}$$

This is because the sensitivity of $\|\nabla f_{\mathcal{D}}(x) - \nabla f_{\mathcal{D}'}(x)\|^2 \leq \frac{S^2}{n^2} \times (2\eta M)^2$ for group size $S \geq 1$. More specifically, there are at most $S$ different pairs of $\nabla f(x; \mathbf{d}_i) - \nabla f(x; \mathbf{d}'_i)$, and for each pair we have $\|\eta\nabla f(x; \mathbf{d}_i) - \eta\nabla f(x; \mathbf{d}'_i)\| \leq 2\eta M$ by triangle inequality and $M$-Lipschitzness. By triangle inequality again, we have $\|\nabla f_{\mathcal{D}}(x) - \nabla f_{\mathcal{D}'}(x)\|^2 \leq (\frac{2\eta SM}{n})^2$. On the other hand, the variance of the added Gaussian noise in this step (from $x_t$ to $x_{t,1}$) is $\eta\sigma^2$. Leveraging the standard result of Gaussian mechanism [22] gives the $\alpha$-Rényi divergence $\frac{4\alpha\eta^2 S^2 M^2/n^2}{2(\sigma^2\eta)} = \frac{2\alpha\eta S^2 M^2}{\sigma^2 n^2}$. Dividing it by $\alpha$ gives the second term in (47).

Then by applying Lemma H.1, we have

$$\frac{D_\alpha(\nu_{t+1}||\nu'_{t+1})}{\alpha} \leq \frac{D_{\alpha'}(\nu_{t,1}||\nu'_{t,1})}{\alpha'}(1 + \frac{\eta\sigma^2}{C_{t,1}})^{-1}, \; \alpha' = \frac{\alpha - 1}{1 + \frac{\eta\sigma^2}{C_{t,1}}} + 1. \tag{48}$$

Combining these two bounds we have

$$\frac{D_\alpha(\nu_{t+1}||\nu'_{t+1})}{\alpha} \le \left(\frac{D_{\alpha'}(\nu_t||\nu'_t)}{\alpha'} + \frac{2\eta S^2 M^2}{\sigma^2 n^2}\right)(1+\frac{\eta\sigma^2}{C_{t,1}})^{-1}, \ \alpha' = \frac{\alpha-1}{1+\frac{\eta\sigma^2}{C_{t,1}}} + 1. \qquad (49)$$

Now, iterate this bound for all $t$ and note that $D_\alpha(\nu_0||\nu'_0) = 0$ for any $\alpha > 1$ due to the same initialization, we have

$$\frac{D_\alpha(\nu_T||\nu'_T)}{\alpha} \le \frac{2\eta S^2 M^2}{\sigma^2 n^2} \sum_{t=1}^{T} \prod_{t'=0}^{t-1} (1+\frac{\eta\sigma^2}{C_{t',1}})^{-1}. \qquad (50)$$

The same analysis applies to the other direction $\frac{D_\alpha(\nu'_{t+1}||\nu_{t+1})}{\alpha}$. Together we complete the proof for convex and non-convex cases. For the $m$-strongly convex case, it is a direct result of Theorem D.6 in [18], where the LSI constant analysis of $C_t$ is exactly the same to those of Theorem 3.2. Together we complete the proof. $\square$

# I  Proof of Corollary 3.4

**Corollary I.1** (Sequential unlearning). *Assume the unlearning requests arrive sequentially such that our dataset changes from $\mathcal{D} = \mathcal{D}_0 \to \mathcal{D}_1 \to \ldots \to \mathcal{D}_S$, where $\mathcal{D}_s, \mathcal{D}_{s+1}$ are adjacent. Let $y_k^{(s)}$ be the unlearned parameters for the $s^{th}$ unlearning request with $k$ unlearning update following (2) on $\mathcal{D}_s$ and $y_0^{(s+1)} = y_{K_s}^{(s)} \sim \bar{\nu}_{\mathcal{D}_s}$, where $y_0^{(1)} = x_\infty$ and $K_s$ is the unlearning steps for the $s^{th}$ unlearning request. Suppose we have achieved $(\alpha, \varepsilon^{(s)}(\alpha))$-RU for the $s^{th}$ unlearning request, the learning process (1) is $(\alpha, \varepsilon_0(\alpha))$-RDP and $\bar{\nu}_{\mathcal{D}_s}$ satisfies $C_{LSI}$-LSI, we achieve $(\alpha, \varepsilon^{(s+1)}(\alpha))$-RU for the $(s+1)^{th}$ unlearning request as well, where*

$$\varepsilon^{(s+1)}(\alpha) \le \exp(-\frac{1}{\alpha}\sum_{k=0}^{K_{s+1}-1} R_k)\frac{\alpha-1/2}{\alpha-1}\left(\varepsilon_0(2\alpha) + \varepsilon^{(s)}(2\alpha)\right),$$

$\varepsilon^{(0)}(\alpha) = 0 \ \forall \alpha > 1$ *and $R_k$ are defined in Theorem 3.2.*

While our main theorems only discuss one unlearning request, we can generalize it to address multiple unlearning requests. Consider the case where our learning process is trained with dataset $\mathcal{D}$. At the unlearning phase, we receive a sequence of unlearning requests so that our dataset becomes $\mathcal{D}_1, \mathcal{D}_2, \ldots, \mathcal{D}_S$, where each consecutive dataset $\mathcal{D}_s, \mathcal{D}_{s+1}$ are adjacent (i.e., each unlearning request ask for unlearning one data point). Let us denote $\nu_{\mathcal{D}_s}$ the output probability distribution of $\mathcal{M}(\mathcal{D}_s)$ for $s \ge 0$, where we set $\mathcal{D}_0 = \mathcal{D}$. Sequential unlearning can be viewed as transferring along $\nu_{\mathcal{D}_0} \to \nu_{\mathcal{D}_1} \cdots \to \nu_{\mathcal{D}_S}$, where for each request we will stop when we are "$\varepsilon$" away from the target distribution in terms of Rényi difference. As a result, our actual path is $\nu_{\mathcal{D}_0} \to \bar{\nu}_{\mathcal{D}_1} \cdots \to \bar{\nu}_{\mathcal{D}_S}$ for some sequence of distribution $\{\bar{\nu}_{\mathcal{D}_s}\}_{s=1}^S$ such that the $\alpha$ Rényi difference $d_\alpha(\nu_{\mathcal{D}_s}, \bar{\nu}_{\mathcal{D}_s}) \le \varepsilon$. See Figure 2 for a pictorial example of the case $S = 2$. While we are unable to characterize the convergence along $\bar{\nu}_{\mathcal{D}_s} \to \bar{\nu}_{\mathcal{D}_{s+1}}$ directly, we can leverage the weak triangle inequality of Rényi divergence to provide an upper bound of it.

**Proposition I.2** (Weak Triangle Inequality of Rényi divergence, Corollary 4 in [22]). *For any $\alpha > 1$, $p, q > 1$ satisfying $1/p + 1/q = 1$ and distributions $P, Q, R$ with the same support:*

$$D_\alpha(P||R) \le \frac{\alpha - \frac{1}{p}}{\alpha - 1}D_{p\alpha}(P||Q) + D_{q(\alpha-1/p)}(Q||R).$$

Note that by choosing $p = q = 2$, we can also establish the weak triangle inequality for Rényi difference $d_\alpha$ as follows

$$D_\alpha(P||R) \leq \frac{\alpha - \frac{1}{2}}{\alpha - 1} D_{2\alpha}(P||Q) + D_{2\alpha-1}(Q||R) \tag{51}$$

$$\overset{(a)}{\leq} \frac{\alpha - \frac{1}{2}}{\alpha - 1} d_{2\alpha}(P, Q) + d_{2\alpha-1}(Q, R) \tag{52}$$

$$\overset{(b)}{\leq} \frac{\alpha - \frac{1}{2}}{\alpha - 1} d_{2\alpha}(P, Q) + d_{2\alpha}(Q, R) \tag{53}$$

$$\overset{(c)}{\leq} \frac{\alpha - \frac{1}{2}}{\alpha - 1} \left( d_{2\alpha}(P, Q) + d_{2\alpha}(Q, R) \right), \tag{54}$$

$$\tag{55}$$

where (a) is due to the definition of Rényi difference, (b) is due to the monotonicity of Rényi divergence in $\alpha$ and (c) is due to the fact that for all $\alpha > 1$, $\frac{\alpha - \frac{1}{2}}{\alpha - 1} \geq 1$. Repeat the same analysis for $D_\alpha(R||P)$ and combine with the bound above, one can show that

$$d_\alpha(P, R) \leq \frac{\alpha - \frac{1}{2}}{\alpha - 1} \left( d_{2\alpha}(P, Q) + d_{2\alpha}(Q, R) \right). \tag{56}$$

The main idea is illustrated in Figure 2 (a). We first leverage Theorem 3.2 to upper bound the Rényi difference $d_\alpha(\bar{\nu}_{\mathcal{D}_2}, \nu_{\mathcal{D}_2})$ in terms of the Rényi difference between $d_\alpha(\bar{\nu}_{\mathcal{D}_1}, \nu_{\mathcal{D}_2})$ (dash line) with a decaying factor. Then by weak triangle inequality of Rényi difference we derived above, we can further bound it with $\varepsilon^{(1)}(2\alpha)$ (black line) and $\varepsilon_0(2\alpha)$ (red line).

*Proof.* The proof is a direct combination of Theorem 3.2 and Proposition I.2. To achieve $(\alpha, \varepsilon^{(s+1)}(\alpha))$-RU for the $(s+1)^{th}$ unlearning request, we need to bound $d_\alpha(\tilde{\nu}_{\mathcal{D}_{s+1}}, \nu_{\mathcal{D}_{s+1}})$. Assume we run $K_{s+1}$ unlearning iteration, from Theorem 3.2 we have

$$d_\alpha(\tilde{\nu}_{\mathcal{D}_{s+1}}, \nu_{\mathcal{D}_{s+1}}) \leq \exp(-\frac{1}{\alpha} \sum_{k=0}^{K_{s+1}-1} R_k) d_\alpha(\tilde{\nu}_{\mathcal{D}_s}, \nu_{\mathcal{D}_{s+1}}), \tag{57}$$

where $R_k$ is defined in Theorem 3.2. On the other hand, by weak triangle inequality of Rényi difference, we have

$$d_\alpha(\tilde{\nu}_{\mathcal{D}_s}, \nu_{\mathcal{D}_{s+1}}) \leq \frac{\alpha - 1/2}{\alpha - 1} \left( d_{2\alpha}(\tilde{\nu}_{\mathcal{D}_s}, \nu_{\mathcal{D}_s}) + d_{2\alpha}(\nu_{\mathcal{D}_s}, \nu_{\mathcal{D}_{s+1}}) \right). \tag{58}$$

By the initial RDP condition, we know that $d_{2\alpha}(\nu_{\mathcal{D}_s}, \nu_{\mathcal{D}_{s+1}}) \leq \varepsilon_0(2\alpha)$. On the other hand, by the RU guarantee of the $s^{th}$ unlearning request, we have

$$d_{2\alpha}(\tilde{\nu}_{\mathcal{D}_s}, \nu_{\mathcal{D}_s}) \leq \varepsilon^{(s)}(2\alpha). \tag{59}$$

Together we have

$$d_\alpha(\tilde{\nu}_{\mathcal{D}_s}, \nu_{\mathcal{D}_{s+1}}) \leq \frac{\alpha - 1/2}{\alpha - 1} \left( \varepsilon_0(2\alpha) + \varepsilon^{(s)}(2\alpha) \right). \tag{60}$$

Hence we complete the proof. $\qquad\square$

## J  Proof of Lemma G.1

**Lemma** (LSI constant characterization). *Consider the following PNGD update for a closed convex set $\mathcal{C}$:*

$$x_{k,1} = h(x_k), \ x_{k,2} = x_{k,1} + \sigma W_k, \ x_{k+1} = \Pi_\mathcal{C}(x_{k,2}),$$

*where $h$ is any $M$-Lipschitz map $\mathbb{R}^d \mapsto \mathbb{R}^d$, $W_k \sim \mathcal{N}(0, I_d)$ independent of anything before step $k$, and $\Pi_\mathcal{C}$ is the projection onto $\mathcal{C}$. Let $\mu_{k,1}, \mu_{k,2}$ and $\mu_k$ be the probability distribution of $x_{k,1}, x_{k,2}$ and $x_k$ respectively. Then we have the following LSI constant characterization of this process. 1) If $\mu_k$ satisfies $c$-LSI, $\mu_{k,1}$ satisfies $M^2 c$-LSI. 2) If $\mu_{k,1}$ satisfies $c$-LSI, $\mu_{k,2}$ satisfies $(c + \sigma^2)$-LSI. 3) If $\mu_{k,2}$ satisfies $c$-LSI, $\mu_{k+1}$ satisfies $c$-LSI.*

*Proof.* The first statement is the direct result of Proposition 2.3.3. in [26]. See also Lemma 16 in [15] but additionally require $h$ being differentiable. The second statement is the direct result of Lemma 17 in [15]. The third statement is because $\Pi_{\mathcal{C}}$ is a 1-Lipchitz map. Together we complete the proof. $\qquad\square$

## K  Proof of Lemma G.4

**Lemma** (Lemma 18 in [15], with customized variance)**.** *For any probability distribution $\rho_0$, $\nu_0$ and for any $t \geq 0$, let $\rho_t = \rho_0 * \mathcal{N}(0, 2t\sigma^2 I_d)$ and $\nu_t = \nu_0 * \mathcal{N}(0, 2t\sigma^2 I_d)$. Then for all $\alpha > 0$ we have*

$$\frac{d}{dt} D_\alpha(\rho_t || \nu_t) = -\alpha\sigma^2 \frac{G_\alpha(\rho_t; \nu_t)}{F_\alpha(\rho_t; \nu_t)}. \tag{61}$$

*Proof.* The proof is nearly identical to that in [15]. Let $X_t \sim \rho_t$, then we have the following stochastic differential equation.

$$dX_t = \sqrt{2}\sigma dW_t. \tag{62}$$

Thus $\rho_t$ evolves following the Fokker-Planck equation:

$$\frac{\partial \rho_t}{\partial t} = \sigma^2 \Delta \rho_t. \tag{63}$$

Same for $\nu_t$ and just plug this into the first step in the proof of Lemma 18 in [15], which gives the result. $\qquad\square$

## L  Proof of Proposition E.1

The proof is a direct manipulation of the Rényi divergence. Due to symmetry, we will only show that $D_\alpha(\tilde{\nu}_{\mathcal{D}}, \tilde{\nu}_{\mathcal{D}'}) \leq \frac{2\alpha F}{(\alpha-1)n}$, as the proof for the bound of $D_\alpha(\tilde{\nu}_{\mathcal{D}'}, \tilde{\nu}_{\mathcal{D}})$ is identical.

Define $Z_{\mathcal{D}} = \int \exp(-f_{\mathcal{D}}(x))dx$ be the normalizing constant. Then we have

$$D_\alpha(\tilde{\nu}_{\mathcal{D}}, \tilde{\nu}_{\mathcal{D}'}) = \frac{1}{\alpha-1} \log \mathbb{E}_{x \sim \tilde{\nu}_{\mathcal{D}'}} \left( \frac{\tilde{\nu}_{\mathcal{D}}(x)}{\tilde{\nu}_{\mathcal{D}'}(x)} \right)^\alpha = \frac{1}{\alpha-1} \log \mathbb{E}_{x \sim \tilde{\nu}_{\mathcal{D}}} \left( \frac{\tilde{\nu}_{\mathcal{D}}(x)}{\tilde{\nu}_{\mathcal{D}'}(x)} \right)^{\alpha-1} \tag{64}$$

$$= \frac{1}{\alpha-1} \log \left( \left( \frac{Z_{\mathcal{D}'}}{Z_{\mathcal{D}}} \right)^{\alpha-1} \mathbb{E}_{x \sim \tilde{\nu}_{\mathcal{D}}} \left( \frac{\exp(-f_{\mathcal{D}}(x))}{\exp(-f_{\mathcal{D}'}(x))} \right)^{\alpha-1} \right) \tag{65}$$

$$= \log(\frac{Z_{\mathcal{D}'}}{Z_{\mathcal{D}}}) + \frac{1}{\alpha-1} \log(\mathbb{E}_{x \sim \tilde{\nu}_{\mathcal{D}}} \left( \frac{\exp(-f_{\mathcal{D}}(x))}{\exp(-f_{\mathcal{D}'}(x))} \right)^{\alpha-1}). \tag{66}$$

Recall that $\mathcal{D}$ and $\mathcal{D}'$ are adjacent, thus they only differ in one index. Without loss of generality, assume the index is $n$ so that $\mathbf{d}_i = \mathbf{d}'_i$ for all $i < n$. By definition,

$$f_{\mathcal{D}'}(x) = \frac{1}{n} \sum_{i=1}^{n-1} f(x; \mathbf{d}'_i) + \frac{1}{n} f(x; \mathbf{d}'_n) \tag{67}$$

$$= \frac{1}{n} \sum_{i=1}^{n-1} f(x; \mathbf{d}'_i) + \frac{1}{n} f(x; \mathbf{d}_n) + \frac{1}{n} f(x; \mathbf{d}'_n) - \frac{1}{n} f(x; \mathbf{d}_n) \tag{68}$$

$$= f_{\mathcal{D}}(x) + \frac{1}{n}(f(x; \mathbf{d}'_n) - f(x; \mathbf{d}_n)). \tag{69}$$

As a result, the ratio of the normalizing constant can be bounded as

$$\frac{Z_{\mathcal{D}'}}{Z_{\mathcal{D}}} = \frac{\int \exp(-f_{\mathcal{D}'}(x))dx}{Z_{\mathcal{D}}} = \frac{\int \exp(-f_{\mathcal{D}'}(x))dx}{Z_{\mathcal{D}}} = \frac{\int \exp(-f_{\mathcal{D}}(x) + \frac{f(x;\mathbf{d}'_n)-f(x;\mathbf{d}_n)}{n})dx}{Z_{\mathcal{D}}} \tag{70}$$

$$\leq \frac{\int \exp(-f_{\mathcal{D}}(x) + \frac{|f(x;\mathbf{d}'_n)-f(x;\mathbf{d}_n)|}{n})dx}{Z_{\mathcal{D}}} \tag{71}$$

$$\leq \frac{\int \exp(-f_{\mathcal{D}}(x) + \frac{F}{n})dx}{Z_{\mathcal{D}}} = \frac{\exp(\frac{F}{n}) \int \exp(-f_{\mathcal{D}}(x))dx}{Z_{\mathcal{D}}} = \frac{\exp(\frac{F}{n})Z_{\mathcal{D}}}{Z_{\mathcal{D}}} = \exp(\frac{F}{n}). \tag{72}$$

On the other hand, for the second term we have

$$\mathbb{E}_{x \sim \tilde{\nu}_{\mathcal{D}}} \left( \frac{\exp(-f_{\mathcal{D}}(x))}{\exp(-f_{\mathcal{D}'}(x))} \right)^{\alpha-1} = \mathbb{E}_{x \sim \tilde{\nu}_{\mathcal{D}}} \exp(-(\alpha-1)(f_{\mathcal{D}}(x) - f_{\mathcal{D}'}(x))) \tag{73}$$

$$\leq \mathbb{E}_{x \sim \tilde{\nu}_{\mathcal{D}}} \exp((\alpha-1)(\frac{F}{n})) = \exp((\alpha-1)(\frac{F}{n})). \tag{74}$$

As a result, we can further simplify (64) as follows

$$D_{\alpha}(\tilde{\nu}_{\mathcal{D}}, \tilde{\nu}_{\mathcal{D}'}) = \log(\frac{Z_{\mathcal{D}'}}{Z_{\mathcal{D}}}) + \frac{1}{\alpha-1} \log(\mathbb{E}_{x \sim \tilde{\nu}_{\mathcal{D}}} \left( \frac{\exp(-f_{\mathcal{D}}(x))}{\exp(-f_{\mathcal{D}'}(x))} \right)^{\alpha-1}) \tag{75}$$

$$\leq \frac{F}{n} + \frac{(\alpha-1)F}{(\alpha-1)n} = \frac{2F}{n}. \tag{76}$$

Together we complete the proof.

# M   Experiment Details

## M.1   $(\alpha, \varepsilon)$-RU to $(\epsilon, \delta)$-Unlearning Conversion

Let us first state the definition of $(\epsilon, \delta)$-unlearning from prior literature [1, 7, 8].

**Definition M.1.** Consider a randomized learning algorithm $\mathcal{M} : \mathcal{X}^n \mapsto \mathbb{R}^d$ and a randomized unlearning algorithm $\mathcal{U} : \mathbb{R}^d \times \mathcal{X}^n \times \mathcal{X}^n \mapsto \mathbb{R}^d$. We say $(\mathcal{M}, \mathcal{U})$ achieves $(\epsilon, \delta)$-unlearning if for any adjacent datasets $\mathcal{D}, \mathcal{D}'$ and any event $E$, we have

$$\mathbb{P}\left(\mathcal{U}(\mathcal{M}(\mathcal{D}), \mathcal{D}, \mathcal{D}') \subseteq E\right) \leq \exp(\epsilon)\mathbb{P}\left(\mathcal{M}(\mathcal{D}') \subseteq E\right) + \delta, \tag{77}$$

$$\mathbb{P}\left(\mathcal{M}(\mathcal{D}') \subseteq E\right) \leq \exp(\epsilon)\mathbb{P}\left(\mathcal{U}(\mathcal{M}(\mathcal{D}), \mathcal{D}, \mathcal{D}') \subseteq E\right) + \delta. \tag{78}$$

Following the same proof of RDP-DP conversion (Proposition 3 in [22]), we have the following $(\alpha, \varepsilon)$-RU to $(\epsilon, \delta)$-unlearning conversion as well.

**Proposition M.2.** *If $(\mathcal{M}, \mathcal{U})$ achieves $(\alpha, \varepsilon)$-RU, it satisfies $(\epsilon, \delta)$-unlearning as well, where*

$$\epsilon = \varepsilon + \frac{\log(1/\delta)}{\alpha-1}. \tag{79}$$

## M.2   Datasets

**MNIST** [34] contains the grey-scale image of number $0$ to number $9$, each with $28 \times 28$ pixels. We follow [8] to take the images with the label $3$ and $8$ as the two classes for logistic regression. The training data contains $11982$ instances in total and the testing data contains $1984$ samples. We spread the image into an $x \in \mathbb{R}^d, d = 724$ feature as the input of logistic regression.

**CIFAR-10** [35] contains the RGB-scale image of ten classes for image classification, each with $32 \times 32$ pixels. For **CIFAR-10-binary**, we also select class #3 (cat) and class #8 (ship) as the two classes for logistic regression. The training data contains $10000$ instances and the testing data contains $2000$ samples. As to **CIFAR-10-multi-class**, we include all the classes for multi-class logistic regression. The training data contains $50000$ instances and the testing data contains $10000$ samples. We apply data pre-processing on CIFAR-10 by extracting the compact feature encoding from the last layer before pooling of an off-the-shelf pre-trained ResNet18 model [36] from Torchvision library [48, 49] as the input of our logistic regression. The compact feature encoding is $x \in \mathbb{R}^d, d = 512$.

All the inputs from the datasets are normalized with the $\ell_2$ norm of $1$.

## M.3   Experiment Settings

**Hardware and Frameworks** All the experiments run with PyTorch=2.1.2 [50] and numpy=1.24.3 [51]. The codes run on a server with a single NVIDIA RTX 6000 GPU with AMD EPYC 7763 64-Core Processor.

**Problem Formulation** Given a binary classification task $\mathcal{D} = \{\mathbf{x}_i \in \mathbb{R}^d, y_i \in \{-1, +1\}\}_{i=1}^n$, our goal is to obtain a set of parameters $\mathbf{w}$ that optimizes the objective below:

$$\mathcal{L}(\mathbf{w}; \mathcal{D}) = \frac{1}{n} \sum_{i=1}^n l(\mathbf{w}^\top \mathbf{x}_i, y_i) + \frac{\lambda}{2} \|\mathbf{w}\|_2^2, \tag{80}$$

where the objective consists of a standard logistic regression loss $l(\mathbf{w}^\top x_i, y_i) = -\log \sigma(y_i \mathbf{w}^\top \mathbf{x}_i)$, where $\sigma(t) = \frac{1}{1+\exp(-t)}$ is the sigmoid function; and a $\ell_2$ regularization term where $\lambda$ is a hyper-parameter to control the regularization, and we set $\lambda$ as $10^{-6} \times n$ across all the experiments. By simple algebra one can show that [7]

$$\nabla l(\mathbf{w}^\top \mathbf{x}_i, y_i) = (\sigma(y_i \mathbf{w}^\top \mathbf{x}_i) - 1) y_i \mathbf{x}_i + \lambda \mathbf{w}, \tag{81}$$

$$\nabla^2 l(\mathbf{w}^\top \mathbf{x}_i, y_i) = \sigma(y_i \mathbf{w}^\top \mathbf{x}_i)(1 - \sigma(y_i \mathbf{w}^\top \mathbf{x}_i)) \mathbf{x}_i \mathbf{x}_i^T + \lambda I_d. \tag{82}$$

Due to $\sigma(y_i \mathbf{w}^\top \mathbf{x}_i) \in [0, 1]$, it is not hard to see that we have smoothness $L = 1/4 + \lambda$ and strong convexity $\lambda$.

The per-sample gradient with clipping w.r.t. the weights $\mathbf{w}$ of the logistic regression loss function is given as:

$$\nabla_{clip} l(\mathbf{w}^\top \mathbf{x}_i, y_i) = \Pi_{\mathcal{C}_M} \left( (\sigma(y_i \mathbf{w}^\top \mathbf{x}_i) - 1) y_i \mathbf{x}_i \right) + \lambda \mathbf{w}, \tag{83}$$

where $\Pi_{\mathcal{C}_M}$ denotes the gradient clipping projection into the Euclidean ball with the radius of $M$, to satisfy the Lipschitz constant bound. According to Proposition 5.2 of [18], the per-sampling clipping operation still results in a $L$-smooth, $m$-strongly convex objective. The resulting Langevin learning/unlearning update on the full dataset is as follows:

$$\frac{1}{n} \sum_{i=1}^n \nabla_{clip} l(\mathbf{w}^T \mathbf{x}_i, y_i), \tag{84}$$

Finally, we remark that in our specific case since we have normalized the features of all data points (i.e., $\|x\| = 1$), by the explicit gradient formula we know that $\|(\sigma(y_i \mathbf{w}^\top \mathbf{x}_i) - 1) y_i \mathbf{x}_i\| \leq 1$.

As to multi-class classification task $\mathcal{D} = \{\mathbf{x}_i \in \mathbb{R}^d, y_i \in \{-1, +1\}^c\}_{i=1}^n$, the loss function is denoted as follows [18]:

$$\mathcal{L}(\mathbf{w}; \mathcal{D}) = \frac{1}{n} \sum_{i=1}^n l(\mathbf{w}^\top \mathbf{x}_i, y_i) + \frac{\lambda}{2} \|\mathbf{w}\|_2^2, \tag{85}$$

where

$$l(\mathbf{w}^\top \mathbf{x}_i, y_i) = -y^1 \log\left(\frac{e^{w_1^\top x_i}}{e^{w_1^\top x_i} + ... + e^{w_c^\top x_i}}\right) - ... - y^c \log\left(\frac{e^{w_c^\top x_i}}{e^{w_1^\top x_i} + ... + e^{w_c^\top x_i}}\right). \tag{86}$$

After similar derivations, the aforementioned objective function can also yield an explicit expression for the gradient, as well as bounds for the constants.

The constant meta-data of the loss function in equation (80) and (85) above for the datasets is shown in the table below:

Table 1: The constants for the loss function and other calculation on MNIST and CIFAR-10.

| | expression | MNIST | CIFAR10-binary | CIFAR10-multi-class |
|---|---|---|---|---|
| smoothness constant $L$ | $\frac{1}{4} + \lambda$ | $\frac{1}{4} + \lambda$ | $\frac{1}{4} + \lambda$ | $1 + \lambda$ |
| strongly convex constant $m$ | $\lambda$ | 0.0119 | 0.0100 | 0.0499 |
| Lipschitz constant $M$ | gradient clip | 1 | 1 | 2 |
| RDP constant $\delta$ | $1/n$ | 8.3458e-5 | 0.0001 | 2e-5 |
| $C_{\text{LSI}}$ | $> \frac{\sigma^2}{m}$ | $\frac{2\sigma^2}{m}$ | $\frac{2\sigma^2}{m}$ | $\frac{2\sigma^2}{m}$ |

**Learning from scratch set-up** For the baselines and our Langevin unlearning framework, we all sample the initial weight $\mathbf{w}$ randomly sampled from i.i.d Gaussian distribution $\mathcal{N}(\mu_0, C_{\text{LSI}})$, where

$\mu_0$ is a hyper-parameter denoting the initialization mean and we set as $1000$ to simulate the situation where the initial $w$ has a long distance towards the optimum alike most situations in real-world applications. For the learning methods $\mathcal{M}$, we set $T = 10,000$ for all the methods to converge.

**Unlearning request implementation.** In our experiment, for an unlearning request of removing data point $i$, we replace its feature with random features drawn from $\mathcal{N}(0, I_d)$ and its label with a random label drawn uniformly at random drawn from all possible classes. This is similar to the DP replacement definition defined in [52], where they replace a point with a special *null* point $\perp$.

### General implementation of baseline D2D [8]

● Across all of our experiments involved with D2D, we follow the original paper to set the step size as $2/(L + m)$.

● For the experiments in Fig. 3a, we calculate the noise to add after gradient descent with the non-private bound as illustrated in Theorem. N.1 (Theorem 9 in [8]); For experiments with sequential unlearning requests in Fig. 3b, we calculate the least step number and corresponding noise with the bound in Theorem. N.2 (Theorem 28 in [8]).

● The implementation of D2D follows the pseudo code shown in Algorithm 1,2 in [8] as follows:

---

**Algorithm 1** D2D: learning from scratch

---

1: **Input**: dataset $D$
2: **Initialize** $\mathbf{w}_0$
3: **for** $t = 1, 2, \ldots, 10000$ **do**
4: $\quad \mathbf{w}_t = \mathbf{w}_{t-1} - \frac{2}{L+m} \times \frac{1}{n} \sum_{i=1}^{n} (\nabla_{clip} l(\mathbf{w}_{t-1}^T \mathbf{x}_i, y_i))$
5: **end for**
6: **Output**: $\hat{\mathbf{w}} = \mathbf{w}_T$

---

**Algorithm 2** D2D: unlearning

---

1: **Input**: dataset $D_{i-1}$, update $u_i$; model $\mathbf{w}_i$
2: **Update dataset** $D_i = D_{i-1} \circ u_i$
3: **Initialize** $\mathbf{w}_0' = \mathbf{w}_i$
4: **for** $t = 1, \ldots, I$ **do**
5: $\quad \mathbf{w}_t' = \mathbf{w}_{t-1}' - \frac{2}{L+m} \times \frac{1}{n} \sum_{i=1}^{n} \nabla_{clip} l((\mathbf{w}_{t-1}')^T \mathbf{x}_i, y_i)$
6: **end for**
7: **Calculate** $\gamma = \frac{L-m}{L+m}$
8: **Draw** $Z \sim \mathcal{N}(0, \sigma^2 I_d)$
9: **Output** $\hat{\mathbf{w}}_i = \mathbf{w}_{T_i}' + Z$

---

The settings and the calculation of $I, \sigma$ in Algorithm. 2 are discussed in the later part of this section and could be found in Section. N.

**General Implementation of Langevin Unlearning**

- We set the step size $\eta$ for Langevin unlearning framework across all the experiments as $1/L$.

- The pseudo code for Langevin unlearning framework is as follows:

---
**Algorithm 3** Langevin unlearning framework, learning / unlearning
---
1: **Input**: dataset $D$
2: **if** Learn from scratch **then**
3:     **Initialize** $\mathbf{w}_0 \in \mathcal{N}(\mu_0, C_{\text{LSI}}I_d)$
4: **else**
5:     **Initialize** $\mathbf{w}_0$ with the pre-trained parameters
6: **end if**
7: **for** $t = 1, 2, \ldots, K$ **do**
8:     **Draw** $W \overset{\text{iid}}{\sim} \mathcal{N}(0, I_d)$
9:     $\mathbf{w}_t = \mathbf{w}_{t-1} - \frac{1}{L} \times \frac{1}{n} \sum_{i=1}^{n} (\nabla_{clip} l(\mathbf{w}_{t-1}^T \mathbf{x}_i, y_i)) + \sqrt{2\frac{\sigma^2}{L}} W$
10: **end for**
11: **Output**: $\hat{\mathbf{w}} = \mathbf{w}_K$

---

## M.4 Implementation Details for Fig. 3a

In this experiment, we first train the methods on the original dataset $\mathcal{D}$ from scratch to obtain the initial weights $\mathbf{w}_0$. Then we randomly remove a single data point ($S = 1$) from the dataset to get the new dataset $\mathcal{D}'$, and unlearn the methods from the initial weights $\hat{\mathbf{w}}$ and test the accuracy on the testing set.

we set the target $\hat{\epsilon}$ with 6 different values as $[0.05, 0.1, 0.5, 1, 2, 5]$. For each target $\hat{\epsilon}$:

- For D2D, we set three different unlearning gradient descent step budgets as $I = 1, 2, 5$, and calculate the corresponding noise to be added to the weight after gradient descent on $\mathcal{D}$ according to Theorem. N.1, where the detailed noise information is shown in the table below:

Table 2: Baseline $\sigma$ details in Fig. 3a

|  |  | 0.05 | 0.1 | 0.5 | 1 | 2 | 5 |
|---|---|---|---|---|---|---|---|
|  | 1 | 59.5184 | 29.7994 | 6.0233 | 3.0504 | 1.5626 | 0.6663 |
| CIFAR-10-binary | 2 | 28.1340 | 14.0859 | 2.8472 | 1.4419 | 0.7386 | 0.3149 |
|  | 5 | 9.4523 | 4.7325 | 0.9565 | 0.4844 | 0.2481 | 0.1058 |
|  | 1 | 5.9612 | 2.9840 | 0.6022 | 0.3044 | 0.1554 | 0.0657 |
| CIFAR-10-multi-class | 2 | 2.8386 | 1.4209 | 0.2867 | 0.1449 | 0.0740 | 0.0313 |
|  | 5 | 0.9764 | 0.4887 | 0.0986 | 0.0498 | 0.0254 | 0.0107 |
|  | 1 | 36.8573 | 18.4620 | 3.7310 | 1.8890 | 0.9673 | 0.4120 |
| MNIST | 2 | 17.3030 | 8.6229 | 1.7507 | 0.8864 | 0.4538 | 0.1933 |
|  | 5 | 5.6774 | 2.8424 | 0.5744 | 0.2908 | 0.1489 | 0.0634 |

- For the Langevin unlearning framework, we set the unlearning fine-tune step budget as $\hat{K} = 1$ only, and calculate the smallest $\sigma$ that could satisfy the fine-tune step budget and target $\hat{\epsilon}$ at the same time. The calculation follows the binary search algorithm as follows:

---

**Algorithm 4** Langevin Unlearning: binary search $\sigma$ that satisfy $\hat{K}$ and target $\hat{\epsilon}$ budget

---

1: **Input**:target $\hat{\epsilon}$, unlearn step budget $K$, lower bound $\sigma_{\text{low}}$, upper bound $\sigma_{\text{high}}$
2: **while** $\sigma_{\text{low}} \leq \sigma_{\text{high}}$ **do**
3:  $\sigma_{\text{mid}} = (\sigma_{\text{low}} + \sigma_{\text{high}})/2$
4:  call Alg. 5 to find the least $K$ that satisfies $\hat{\epsilon}$ with $\sigma = \sigma_{\text{mid}}$
5:  **if** $K == \hat{K}$ **then**
6:   **Return** $K$
7:  **else if** $K \leq \hat{K}$ **then**
8:   $\sigma_{\text{high}} = \sigma_{\text{mid}}$
9:  **else**
10:   $\sigma_{\text{low}} = \sigma_{\text{mid}}$
11:  **end if**
12: **end while**

---

---

**Algorithm 5** Langevin Unlearning: find the least unlearn step $K$ that satisfies the target $\hat{\epsilon}$

---

1: **Input**:target $\hat{\epsilon}$, $\sigma$
2: **Initialize** $K = 1, \epsilon > \hat{\epsilon}$
3: **while** $\epsilon > \hat{\epsilon}$ **do**
4:  $\epsilon = \min_{\alpha>1}[\exp(-\frac{2K\sigma^2\eta}{\alpha C_{LSI}})\frac{4\alpha S^2 M^2}{m\sigma^2 n^2} + \frac{\log(\frac{1}{\delta})}{\alpha-1}]$
5:  $K = K + 1$
6: **end while**
7: **Return** $K$

---

The $\sigma$ found is reported in the table below:

Table 3: The $\sigma$ found with different target $\hat{\epsilon}$

| $\hat{\epsilon}$ | 0.05 | 0.1 | 0.5 | 1 | 2 | 5 |
|---|---|---|---|---|---|---|
| CIFAR-10-binary | 0.2431 | 0.1220 | 0.0250 | 0.0125 | 0.0064 | 0.0028 |
| CIFAR-10-multi-class | 0.0473 | 0.0238 | 0.0049 | 0.0025 | 0.0012 | 0.0005 |
| MNIST | 0.1872 | 0.094 | 0.0190 | 0.0096 | 0.0049 | 0.0021 |

### M.5 Implementation Details for Fig. 3b

In this experiment, we fix the target $\hat{\epsilon} = 1$, we set the total number of data removal as 100. We show the accumulated unlearning steps w.r.t. the number of data removed. We first train the methods from scratch to get the initial weight $\mathbf{w}_0$, and sequentially remove data step by step until all the data points are removed. We count the accumulated unlearning steps $K$ needed in the process.

• For D2D, According to the original paper, only one data point could be removed a time. We calculate the least required steps and the noise to be added according to Theorem. N.2.

• For Langevin unlearning, we fix the $\sigma = 0.03$, and we let the model unlearn $[5, 10, 20]$ per time thanks to our theory. We obtain the least required unlearning steps for each removal operation $K_{\text{list}}$ following corollary. 3.4. The pseudo code is shown in Algorithm. 6.

---

**Algorithm 6** Langevin Unlearning: find the least unlearn step $K$ in sequential settings

---

1: **Input**:target $\hat{\epsilon}$, $\sigma$, total removal $S$, removal batch size $b$ per time
2: $K_{\text{list}} = []$
3: **for** i in range($S/b$) **do**
4:     **Initialize** $K_{\text{list}}[i-1] = 1$, $\epsilon > \hat{\epsilon}$
5:     **while** $\epsilon > \hat{\epsilon}$ **do**
6:         $\epsilon = \min_{\alpha>1}[\varepsilon(\alpha, \sigma, b, i, K_{\text{list}}) + \log(1/\delta)/(\alpha - 1)]$
7:         $K_{\text{list}}[i-1] = K_{\text{list}}[i-1] + 1$
8:     **end while**
9: **end for**
10: **Return** $K_{\text{list}}$

---

---

**Algorithm 7** $\varepsilon(\alpha, \sigma, b, i, K_{\text{list}})$

---

1: **Input**:target $\alpha$, $\sigma$, removal batch size $b$ per time, $i$-th removal in the sequence
2: **if** i==1 **then**
3:     **Return** $\exp(-\frac{\eta m K_{\text{list}}[0]}{\alpha}) \times \varepsilon_0(\alpha, b, \sigma)$
4: **else**
5:     **Return** $\exp(-\frac{\eta m K_{\text{list}}[i-1]}{\alpha}) \times \frac{\alpha-0.5}{\alpha-1}(\varepsilon_0(2\alpha, b, \sigma) + \varepsilon(2\alpha, \sigma, b, i-1, K_{\text{list}}))$
6: **end if**

---

---

**Algorithm 8** $\epsilon_0(\alpha, S, \sigma)$

---

1: **Return** $\frac{4\alpha S^2 M^2}{m\sigma^2 n^2}$

---

## M.6 Implementation Details for Fig. 3c

In this study, we set the $\sigma$ of the Langevin unlearning framework as $[0.05, 0.1, 0.2, 0.5, 1]$. For each $\sigma$, we calculate the corresponding $\epsilon_0$. We train the Langevin unlearning framework from scratch to get the initial weight $\mathbf{w}_0$. Then we remove 100 data points from the dataset and unlearn the model. We here also call Algorithm. 5 to obtain the least required unlearning steps $K$.

## M.7 Implementation Details for Fig. 4

In this study, we set different target $\hat{\epsilon}$ as $[0.5, 1, 2, 5]$ and set different number of data to remove $S = [1, 50, 100]$. We train the Langevin unlearning framework from scratch to get the initial weight, then remove some data, unlearn the model and report the accuracy. We calculate the least required unlearning steps $K$ by again calling Algorithm. 5.

## M.8 Additional experiments

# N Unlearning Guarantee of Delete-to-Descent [8]

**Theorem N.1** (Theorem 9 in [8], with internal non-private state)**.** *Assume for all $\mathbf{d} \in \mathcal{X}$, $f(x; \mathbf{d})$ is $m$-strongly convex, $M$-Lipschitz and $L$-smooth in $x$. Define $\gamma = \frac{L-m}{L+m}$ and $\eta = \frac{2}{L+m}$. Let the learning iteration $T \geq I + \log(\frac{2Rmn}{2M})/\log(1/\gamma)$ for PGD (Algorithm 1 in [8]) and the unlearning algorithm (Algorithm 2 in [8], PGD fine-tuning on learned parameters **before** adding Gaussian noise) run with $I$ iterations. Assume $\epsilon = O(\log(1/\delta))$, let the standard deviation of the output perturbation gaussian noise $\sigma$ to be*

$$\sigma = \frac{4\sqrt{2}M\gamma^I}{mn(1-\gamma^I)(\sqrt{\log(1/\delta)+\epsilon} - \sqrt{\log(1/\delta)})}. \tag{87}$$

*Then it achieves $(\epsilon, \delta)$-unlearning for add/remove dataset adjacency.*

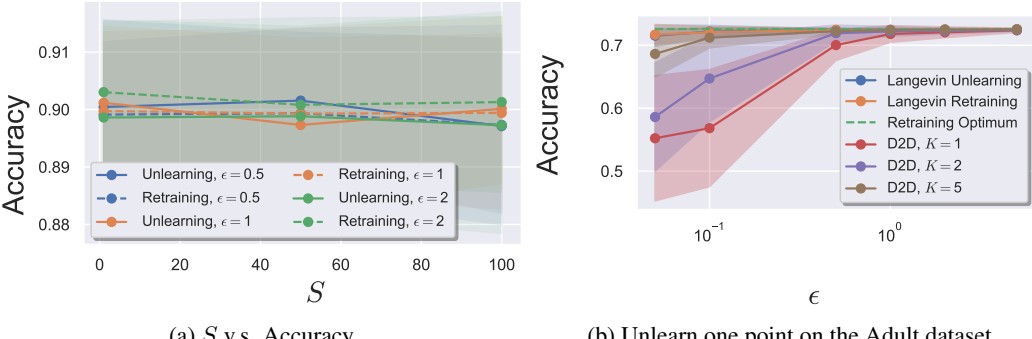

(a) $S$ v.s. Accuracy

(b) Unlearn one point on the Adult dataset

Figure 5: (a) The utility results that correspond to Figure 4. Since $\sigma$ is fixed the utility is roughly the same. (b) The privacy-utility tradeoff for unlearning one point restricting to one (or $K$) unlearning update on the Adult dataset.

**Theorem N.2** (Theorem 28 in [8], without internal non-private state). *Assume for all $\mathbf{d} \in \mathcal{X}$, $f(x; \mathbf{d})$ is $m$-strongly convex, $M$-Lipschitz and $L$-smooth in $x$. Define $\gamma = \frac{L-m}{L+m}$ and $\eta = \frac{2}{L+m}$. Let the learning iteration $T \geq I + \log(\frac{2Rmn}{2M})/\log(1/\gamma)$ for PGD (Algorithm 1 in [8]) and the unlearning algorithm (Algorithm 2 in [8], PGD fine-tuning on learned parameters **after** adding Gaussian noise) run with $I + \log(\log(4di/\delta))/\log(1/\gamma)$ iterations for the $i^{th}$ sequential unlearning request, where $I$ satisfies*

$$I \geq \frac{\log\left(\frac{\sqrt{2d}(1-\gamma)^{-1}}{\sqrt{2\log(2/\delta)+\epsilon}-\sqrt{2\log(2/\delta)}}\right)}{\log(1/\gamma)}. \tag{88}$$

*Assume $\epsilon = O(\log(1/\delta))$, let the standard deviation of the output perturbation gaussian noise $\sigma$ to be*

$$\sigma = \frac{8M\gamma^I}{mn(1-\gamma^I)(\sqrt{2\log(2/\delta)+3\epsilon}-\sqrt{2\log(2/\delta)+2\epsilon})}. \tag{89}$$

*Then it achieves $(\epsilon, \delta)$-unlearning for add/remove dataset adjacency.*

Note that the privacy guarantee of D2D [8] is with respect to add/remove dataset adjacency and ours is the replacement dataset adjacency. However, by a slight modification of the proof of Theorem N.1 and N.2, one can show that a similar (but slightly worse) bound of the theorem above also holds for D2D [8]. For simplicity and fair comparison, we directly use the bound in Theorem N.1 and N.2 in our experiment. Note that [52] also compares a special replacement DP with standard add/remove DP, where a data point can only be replaced with a *null* element in their definition. In contrast, our replacement data adjacency allows *arbitrary* replacement which intuitively provides a stronger privacy notion.

**The non-private internal state of D2D.** There are two different versions of the D2D algorithm depending on whether one allows the server (model holder) to save and leverage the model parameter *before* adding Gaussian noise. The main difference between Theorem N.1 and N.2 is whether their unlearning process starts with the "clean" model parameter (Theorem N.1) or the noisy model parameter (Theorem N.2). Clearly, allowing the server to keep and leverage the non-private internal state provides a weaker notion of privacy [8]. In contrast, our Langevin unlearning approach by default only keeps the noisy parameter so that we do not save any non-private internal state. As a result, one should compare Langevin unlearning to D2D with Theorem N.2 for a fair comparison.

