# OpenReview forum: "Langevin Unlearning: A New Perspective of Noisy Gradient Descent for Machine Unlearning"
_NeurIPS.cc/2024/Conference — NeurIPS 2024 spotlight_

### Official Review · Reviewer_Akdu · 2024-07-09

**Soundness:** 4
**Presentation:** 3
**Contribution:** 3
**Rating:** 8
**Confidence:** 3

**Summary:**

This paper proposes an unlearning framework based on Langevin dynamics with two key components:
- The initially trained model satisfies some differential privacy guarantee
- Unlearning relies on DP fine-tuning the model on the rest of the dataset (which quickly reduces the privacy loss on the example to be forgotten).

This framework is formalized using the theory of convergence of Langevin dynamics. Specifically, assuming the Log-Sobolev inequality holds for the stationary distribution, the paper establishes a linear rate of convergence of the unlearning step. Notably, these bounds apply for the nonconvex setting as well; however, the constants are quite pessimistic in this case. The explicit constants are given for the convex and strongly convex settings, leading to implementable algorithms.

The paper also gives extensions to multiple deletions (sequential and simultaneous). Advantages relative the D2D baseline and retraining from scratch are established as well.

**Strengths:**

This is a very strong paper and generally high quality work.
- While it is grounded in RDP, the theoretical advances are original. I love that the paper also explores 2nd order effects of the proposed framework, such as batch/parallel unlearning, etc.
- This framework gives the first provable approximate unlearning guarantees in the nonconvex setting and improves upon other approaches in the (strongly) convex setting. I believe these results are very significant and can inspire future work in the area.
- The paper is generally well-organized and well-written (although it is not perfect; specific suggestions for improvements are given below).

**Weaknesses:**

- **Clarity**: The paper tackles some advanced technical tools, which may only be known to a niche audience. It would greatly help the broader appeal of the paper to explain the key intuition behind the LSI and why it can be expected to hold. Similarly, it would be nice to consider simple examples in the unlearning setting and give explicit values of the LSI constant. Also, what is its right scaling with respect to other problem parameters such as smoothness, strong convexity, etc.?

- The improvements from Langevin unlearning over training from scratch are purely logarithmic. How significant is this factor in the experiments?

- The paper uses full gradient steps. How do things change with stochastic gradient steps? What would be the computational complexity of the resulting algorithm?

- No experiments in the nonconvex case. Since these are the only provable guarantees in the literature so far, future work in the area would greatly benefit from setting up a benchmark with the results of this work.

**Questions:**

- What are the various degrees of freedom in the privacy-utility-compute tradeoff? I assume that if we fix $\epsilon$ (privacy) and $K$ (compute), then we can tune $\sigma$ to get the desired privacy bound. Are there other degrees of freedom? If there is an extra degree of freedom, what are some rules of thumb for setting the parameters?

- Continuiung from the above, it would be nice to see some more plots exploring this tradeoff. Examples:
     - Fix $K$, plot accuracy vs $\epsilon$ (or vs. $\sigma$) (I believe Fig 3c varies $K$ between the various points)
     - Fix $\sigma$, plot accuracy vs. $\epsilon$ (or vs. $K$)
     - Fix $\epsilon$, plot accuracy vs. $K$

-  How do $C_k$ and $R_k$ scale with $k$ in the various settings of Theorem 3.2? While the authors acknowledge that the rates in the nonconvex case are pessimistic, it would be nice to discuss exactly how the iteration independent lower-bound scales with various problem parameters. Specifically, the dependence on which parameters do you think can be improved?

- Figure 3b: do all of the points have the same utility?

- Theorem 3.2(c): The strongly convex case needs $\sigma^2$ small enough. I wonder about the implications of this requirement. Does it mean that unlearning is not possible using the given framework for certain parameter values? Does it mean that a certain minimum amount of compute is necessary to make unlearning possible? It would be great to explore these quantitatively from the established bounds.

- Can the established unlearning guarantees be compared analytically to existing ones?

**Limitations:**

Yes, limitations have been adequately addressed.

---

> ### Author Rebuttal · Authors · 2024-08-04
>
> We are glad that Reviewer Akdu recognizes our work as a very strong and high-quality paper. We also appreciate the thoughtful comments and suggestions. We address the weaknesses and questions below.
>
> **W1: “Question about LSI constant.”**
>
> Thank you for your valuable suggestion. We will add some intuition about the Log-Sobolev Inequality (LSI) in our revision. Roughly speaking, LSI characterizes the tail density as “well-behaved.” For instance, the density of the form $\nu \propto \exp(-V(x))$ satisfies LSI when the tail behaves as $V(x) \approx \|x\|^2$. If the tail exponent is smaller, say $\|x\|^\alpha$ with $\alpha \in (0,2)$, it corresponds to different isoperimetric inequalities. For a more detailed discussion, please refer to our Appendix B, as well as references [31] and the comprehensive book [26]. See also the section 3 and 4.2 of [31] for some LSI constant example of non-convex distrubiton.
>
> **W2: “How significant is the benefit compared to retraining in the experiments?”**
>
> The significance of the benefit compared to retraining can be observed from Figure 3(a) in the experiment section. In our experiments, Langevin unlearning uses only **one** unlearning update per removed data point, achieving a strong privacy guarantee (i.e., $\epsilon=1$) while maintaining similar accuracy (utility) to retraining from scratch, even when retraining is performed without noise (Retraining Optimum). In contrast, retraining would require **at least** tens of iterations even with a good initialization, and more iterations when the initialization is far from the optimum. This clearly demonstrates that the benefit of our approach compared to retraining is significant in terms of both efficiency and computational cost.
>
> **W3: “Extension to mini-batch setting”**
>
> This is a great question! We have discussed how to correctly extend Langevin unlearning to a mini-batch setting in Appendix A. A more thorough study of this extension is left for future work. The extension to mini-batch settings offers potential improvements in both computational efficiency and adaptability to larger datasets, making it a promising area for further investigation.
>
>
> **Q1: “The privacy-utility-computation trade-off.”**
>
> This is an excellent question! Indeed, for a given $\varepsilon$ (privacy), by fixing $K$ (unlearning steps), we can compute the smallest possible $\sigma$ (noise) to achieve the best possible model utility. Alternatively, another trade-off can be considered: fixing $\sigma$ based on the model utility that one is satisfied with, and then computing the required $K$ that needs to be executed. These approaches allow for a flexible adjustment of parameters based on the desired balance between privacy, utility, and computational cost.
> Experiments to examine the privacy-utility-computation trade-off are indeed crucial, and this is the main focus of Section 4 in our manuscript. Our Figure 3(a) illustrates the trade-off between $\epsilon$ (privacy) and model utility (accuracy) for all tested methods, with $K=1$ fixed unless otherwise specified. Figure 4 further demonstrates the trade-off between $K$ and $\epsilon$ for a fixed $\sigma$. Additionally, Figure 3(c) can be interpreted as a trade-off between accuracy (y-axis on the right, inversely related to $\sigma$) and $K$ for a fixed $\epsilon$. We will provide more explicit explanations and highlight these trade-offs in our revision to ensure clarity for readers.
>
>
> **Q2: “Questions about $C_k$ and $R_k$.”**
>
> To elaborate, $C_k$ scales linearly with $k$ (for the convex case) until it reaches an iteration-independent upper bound $\tilde{C}$, and $R_k$ is roughly the difference $1/C_k^2 - 1/C_{k+1}^2$. For the non-convex case, the dependency on $k$ is worse due to the multiplicative factor $(1+\eta L) > 1$ for $L$-smooth functions and when the step size $\eta > 0$. We believe that the dependencies on all parameters ($L, \eta, k, R$, etc.) can be improved.
>
> Consider a special case where our target distribution is exactly Gaussian $N(0, C I_d)$ with variance $C$. In this scenario, the corresponding LSI constant is $C$, but our bound provides a relatively poor estimate when $C$ is moderate. Intuitively, for sufficiently large $k$, the underlying distribution $\rho_k$ or $\nu_t$ will be close to $N(0, C I_d)$, and the tight bound should be roughly $C$, which is not adequately captured by our Theorem 3.2.
>
> Combined with the discussion in W1, we conjecture that for losses that grow sufficiently fast as the norm of the weight goes to infinity, it may be possible to derive a much tighter estimate of the LSI constant along the (un)learning process.
>
> **Q3: “Do all points in Figure 3(b) have similar utility?”**
>
> Yes. As stated in lines 359-361, all points have similar utility (accuracy) of approximately 0.9 for MNIST and 0.98 for CIFAR-10, respectively.
>
> **Q4: “Question about the strongly convex case of Theorem 3.2.”**
>
> There might be some confusion here. As explained in the simple example provided in W1, it is not necessary to enforce $\sigma$ to be particularly small. Our statement in line 243 pertains purely to the initialization. Note that if a distribution satisfies $C$-LSI, it also satisfies $C^\prime$-LSI for all $C^\prime \geq C$. As a result, given $\sigma^2,m$, we can always choose $C_{LSI}$ to be larger than $\sigma^2/m$, given that the initialization is (roughly) a constant (delta measure) which satisfies $C$-LSI for any $C>0$.
>
> **Q5: “Can the established unlearning guarantees be compared analytically to existing ones?”**
>
> This is a great question. While we can compare our unlearning bounds to existing results in the literature numerically, it is challenging to make an analytical comparison due to the RDP-to-DP conversion process [21]. This complexity is similar in the differential privacy (DP) literature, where researchers typically compare Rényi Differential Privacy (RDP) bounds with DP bounds numerically via such conversions.

---

> > ### Comment · Reviewer_Akdu · 2024-08-09
> > **Thanks!**
> >
> > Thank you for the response! I will keep my score. All the best to the authors!

---

### Official Review · Reviewer_GDGF · 2024-07-12

**Soundness:** 3
**Presentation:** 2
**Contribution:** 3
**Rating:** 5
**Confidence:** 3

**Summary:**

This paper focuses on approximate unlearning under a similar definition of Differential privacy (DP), where privacy is defined as statistical indistinguishability to retraining from scratch. To be specific, they propose Langevin unlearning, which is a new framework based on the noisy gradient descent with privacy guarantees for approximate unlearning. The proposed framework has demonstrated many benefits like approximate certified unlearning for non-convex problems, and sequential and batch unlearning for multiple unlearning requests.

**Strengths:**

1. This work studies the theoretical unlearning guarantees of projected noisy gradient descent algorithms for convex problems, which provide fruitful theoretical basics for unlearning with several important algorithmic benefits, like approximate certified unlearning for the non-convex problems, and complexity saving, as well as enabling the sequential and batch unlearning.
2. The proposed Langevin unlearning based on the projected noisy gradient descent provides a unified geometric view of learning and unlearning processes, which is good for the understanding of the main idea.
3. Both theoretical results and empirical evaluation of the logistic regression task are provided to make the work comprehensive and insightful.

**Weaknesses:**

Overall, the reviewer appreciates the idea of Langevin unlearning with the rigorous theoretical analysis for privacy guarantees, below are some comments from the weakness part which are hoped to be constructive for consideration.

1. The current presentation of this work can be further improved by considering clearly stating all the notations at the corresponding position and reorganizing the theoretical results with some intuitive explanation and connection with the empirical evaluation or justifications.
2. Before introducing the Langevin unlearning with the main results, it could be better to explain the motivation for the proposed framework.
3. For the geometric interpretation of the relations between learning and unlearning, what is the connection between this and the later illustration of the sequential unlearning and batch unlearning scenarios, and could the authors also explain the gap between the illustration and the practical learning dynamics for the corresponding problems?
4. For the empirical aspects of Langevin unlearning, how can we understand the corollary 3.4 more intuitively for the benefit obtained by the framework?

**Questions:**

Minor comments and specific questions:
1. It seems there is no conclusion part, it would be better if the author could summarize the main claims and add it.

**Limitations:**

This work has adequately discussed the limitations and there is no significant negative societal impact of this work.

---

> ### Author Rebuttal · Authors · 2024-08-04
>
> We thank reviewer GDGF for their constructive suggestions and positive assessment. We address the weaknesses and questions below.
>
> **W1: “Suggestions for improving the presentation.”**
>
> We appreciate the helpful suggestions for improving the presentation of our manuscript. Following the suggestion of reviewer GDGF, we plan to add a more intuitive explanation and motivation in Section 3.1, add connections to empirical justification in Sections 3.2-3.3, and improve the clarity of notations.
>
> **W2: “Question about the geometric interpretation for the sequential unlearning and batch unlearning scenarios, and practical learning dynamics.”**
>
> As mentioned in Section 3.1, the learning process can be conceptualized as inducing a regular polyhedron in the space of model weight distribution $\nu_{\mathcal{D}}$. Each vertex represents a model weight distribution $\nu_{\mathcal{D}}$, and adjacent vertices correspond to distributions arising from adjacent datasets $\mathcal{D}$ and $\mathcal{D}^\prime$. The edge length between vertices is indicated by the initial privacy loss $\varepsilon_0$. The unlearning process effectively involves moving from one vertex $\nu_{\mathcal{D}}$ to its adjacent vertex $\nu_{\mathcal{D}^\prime}$ until we are $\varepsilon$-close to the desired distribution.
>
> Within this framework, sequential unlearning is interpreted as moving towards different neighboring vertices in sequence (i.e., $\nu\_{\mathcal{D}} \rightarrow \nu\_{\mathcal{D}\_1} \rightarrow \nu\_{\mathcal{D}\_2}...$), as depicted in Figure 2. It is important to highlight that the unlearning process halts once we are $\varepsilon$-close to the current target distribution. Consequently, the "initial distance" for the subsequent unlearning request is not merely $\varepsilon_0$. Utilizing our geometric interpretation, we apply the weak triangle inequality to provide an upper bound for this initial distance in terms of $\varepsilon_0$ and $\varepsilon$. This allows us to employ Theorem 3.2 to ensure the desired unlearning guarantee.
>
> In the context of batch unlearning, the notion of data adjacency is altered so that datasets $\mathcal{D}$ and $\mathcal{D}^\prime$ are considered adjacent if they differ in $S$ points. This modifies the underlying regular polyhedron, where the edge length (initial distance) now becomes $\varepsilon_0^{(S)}$. Once $\varepsilon_0^{(S)}$ is determined, Theorem 3.2 can again be applied to guarantee successful unlearning.
>
> It is also essential to note that the aforementioned discussion pertains to the worst-case scenario. In practical situations, adjacent (un)learning processes are typically closer in terms of distribution. Recent evidence supporting this claim includes a study [38] demonstrating that models trained using DP-SGD with $\varepsilon \approx 10^{8}$ perform optimally against empirical adversarial attacks, such as membership inference attacks, compared to other popular heuristic defenses. We expect similar behavior to persist for Langevin unlearning, highlighting the practical potential of our approach.
>
> **W3: “Understand the benefit of Corollary 3.4 more intuitively.”**
>
> The primary benefit of Corollary 3.4 is that it facilitates Langevin unlearning for sequential unlearning requests. Sequential unlearning holds significant practical value, particularly in light of regulations like the GDPR that explicitly state, “the data has to be erased without undue delay.” While we do not assert that sequential unlearning is superior to batch unlearning, it is crucial to recognize that these two approaches are orthogonal and can be effectively combined. For instance, one could sequentially unlearn $S\times T$ points by removing $S$ points in each iteration of the process.
>
> **Q1: “No conclusion part”**
>
> We acknowledge reviewer GDGF's suggestion and appreciate the feedback. We will incorporate a conclusion section in the manuscript to summarize our key results and contributions.

---

### Official Review · Reviewer_tZ8F · 2024-07-13

**Soundness:** 3
**Presentation:** 2
**Contribution:** 3
**Rating:** 5
**Confidence:** 1

**Summary:**

This paper proposes a novel framework for machine unlearning through noisy gradient descent with Langevin dynamic analysis. This framework has privacy guarantees for approximate unlearning problems and it unifies differential privacy and machine unlearning process, giving benefits including approximate certified unlearning for non-convex problems, complexity saving compared to retraining, sequential and batch unlearning for multiple unlearning requests.

**Strengths:**

1. The authors propose a novel unlearning framework called Langevin unlearning, which is based on noisy gradient descent. This framework provides a new approach to approximate machine unlearning with privacy guarantees.
2. The paper presents a theory that unifies the differential privacy (DP) learning process with the privacy-certified unlearning process, offering algorithmic benefits such as approximate certified unlearning for non-convex problems.
3. This paper provides empirical results showing the practical effectiveness of the proposed framework compared to existing methods.

**Weaknesses:**

1. The writing is not easy for readers unfamiliar with differential privacy to follow. I think including more details and intuitive explanations in the preliminary would help.
2. Although the paper discusses the computational benefits over retraining, the empirical results are limited to toy datasets such as MNIST and CIFAR10. Actual scalability and computational complexity of the Langevin unlearning approach for large-scale machine learning models and datasets need further investigation.
3. The effect of the method is sensitive to hyperparameters such as the standard deviation of the noise.

**Questions:**

How does the unlearning process ensure other learned data are not affected? Are there any empirical results or theoretical guarantees?

**Limitations:**

While the paper claims to theoretically handle non-convex problems, the practical applicability of the framework for such problems might be limited due to the reliance on certain constants that are either undetermined or can only be loosely determined. The applications may still limited to strong convex problem.

---

> ### Author Rebuttal · Authors · 2024-08-04
>
> We thank reviewer tZ8F for their helpful comments and positive evaluation. We address the weaknesses and questions below.
>
> **W1: “The writing is not easy for readers unfamiliar with differential privacy to follow.”**
>
> We appreciate reviewer tZ8F's insight on making our manuscript more accessible to a broader audience, including those outside the differential privacy (DP) community. To address this concern, we will incorporate more intuitive explanations into Figure 1, Sections 2.1, and 3.1 to make the concepts more approachable. This will help readers unfamiliar with DP better understand our work.
>
> **W2: “The effect of the method is sensitive to hyperparameters such as the standard deviation of the noise.”**
>
> It is important to note that the noise scale (standard deviation $\sigma$) is not a hyperparameter to be tuned arbitrarily. In both DP and theoretical unlearning literature, a privacy constraint $(\epsilon,\delta)$ is specified either by regulatory policies or user agreements. We use our theoretical results (specifically, Theorem 3.2) to determine the smallest possible $\sigma$ that satisfies the given privacy constraint while maintaining the model's utility as much as possible. Therefore, there is no “sensitivity issue” regarding the noise scale, as our theoretical framework provides a privacy-utility trade-off for any given privacy constraint under Definition 2.4.
>
> **Q1: “How does the unlearning process ensure other learned data are not affected? Are there any empirical results or theoretical guarantees?”**
>
> Our Theorem 3.2 provides a theoretical guarantee that the model weight distribution after the unlearning process will be approximately the same as that obtained from retraining the model from scratch without the data that needs to be removed. This essentially means that the other learned data are approximately unaffected by the unlearning process. The gold standard in machine unlearning is retraining from scratch, and our approach closely aligns with this standard by ensuring minimal impact on the remaining data.

---

> > ### Comment · Reviewer_tZ8F · 2024-08-12
> >
> > Thank you for your detailed explanations while I think more empirical results would make the claims more convincing. I will maintain my scores.

---

### Official Review · Reviewer_1Nom · 2024-07-19

**Soundness:** 4
**Presentation:** 4
**Contribution:** 4
**Rating:** 7
**Confidence:** 4

**Summary:**

The paper proposes Langevin unlearning, a new perspective of noisy gradient descent for approximate machine unlearning, which unifies the differential privacy learning process and privacy-certified unlearning process with many algorithmic benefits. The key technique of the proposed method is to carefully track the constant of log-Sobolev inequality. The authors validate the practicality of Langevin unlearning through experiments on MNIST and CIFAR10, and demonstrate its superiority against gradient-descent-plus-output-perturbation based approximate unlearning.

**Strengths:**

1. The authors innovatively interpret the relations between learning and unlearning via a geometric view.

2. The authors provide a rigorous theoretical analysis of privacy guarantees and a framework for certified non-convex approximate unlearning.

3. The theorems and proofs are well-presented.

4. The proposed method supports insufficient training as well as sequential and batch unlearning, aligning well with practical scenarios.

5. The paper is well-organized and the writing is clear overall.

**Weaknesses:**

1. The experiments are not extensive enough and do not consider non-convex problems.

**Questions:**

1. Is it possible to provide experiments on (toy) non-convex problems?

**Limitations:**

1. Most experiments are binary classification problems, with multiclass classification experiments on CIFAR-10-multi-class deferred to the appendix.

---

> ### Author Rebuttal · Authors · 2024-08-04
>
> We thank reviewer 1Nom for their praise and positive feedback. We address their sole question below.
>
> **Q1: “Is it possible to provide experiments on (toy) non-convex problems?”**
>
> Indeed, as mentioned in Section 1.1, the current theoretical bound might not be sufficiently tight for non-convex problems. However, it is possible to conduct experiments on toy non-convex problems such as logistic regression with non-convex regularization, including the minimax concave penalty (MCP) or the smoothly clipped absolute deviation (SCAD) penalty. A canonical toy example of a non-convex function is provided in [31]: $f(x) = \frac{1}{2}\log(1+\|x\|^2) + \frac{1}{20}\|x\|^2$, which is related to Student’s t-regression with a Gaussian prior.
>
> Literature indicates that the distribution $\nu(x) \propto \exp(-f(x))$ satisfies the Log-Sobolev Inequality (LSI) with a constant roughly around $176$ [31]. Also, it is known that the Langevin diffusion process with $f(x)$ as its potential function and noise variance $1$ has a stationary distribution $\nu(x)$, also known as the Gibbs measure. Furthermore, $f(x)$ is known to be $1.1$-Lipschitz and $1.6$-smooth within the $\ell_2$ ball of radius $R=1$.
>
> When the step size $\eta = 1$ and $\sigma = 1$, the iteration-independent LSI constant bound in Theorem 3.2 approximately evaluates to $8 \times 10^5$, which is a significant overestimation and results in impractical privacy bounds. This highlights the reasons we did not conduct non-convex experiments.
>
> Improving the unlearning analysis for non-convex problems, particularly a better estimate of the LSI constant, remains an interesting and significant future direction, as already mentioned in our outlined future work.

---

> > ### Comment · Reviewer_1Nom · 2024-08-12
> > **Thank you for your response**
> >
> > Thank you for your response. I will maintain my score.

---

### Decision · Program_Chairs · 2024-09-25

**Decision:**

Accept (spotlight)

**Comment:**

The paper considers the problem of machine unlearning, which has gained interest in recent years. It proposes a new algorithm, where the initial model is trained with a DP algorithm and then unlearning relies on fine tuning the model on the rest of the dataset, again with differential privacy. Theoretical guarantees are provided for convex, strongly convex, and non-convex settings, and the algorithm is empirically demonstrated on a few small datasets.

While the reviewers are generally positive, they do raise a few concerns: lack of experiments on non-convex problems, theoretical guarantees require full-gradient steps at each iteration, and certain parts of the paper are difficult to follow.  While I agree with the reviewers, I do believe the paper provides a new algorithm and theoretical results that would be of interest to many in the community. I recommend acceptance. I strongly encourage authors to address reviewer concerns in the final version of the paper.